



# Insights from a new methodology to optimize rain gauge weighting for rainfall-runoff models

Ashley J. Wright[1], David E. Robertson[2], Jeffrey P. Walker[1], and Valentijn R.N. Pauwels[1]

[1]Department of Civil Engineering, Monash University, Clayton, Victoria, Australia
[2]Commonwealth Scientific and Industrial Research Organisation, Clayton, Victoria, Australia

**Correspondence:** Ashley Wright          ashley.wright@monash.edu

**Abstract.** Floods continue to devastate societies and their economies. Resilient societies commonly incorporate flood forecasting into their strategy to mitigate the impact of floods. Hydrological models which simulate the rainfall-runoff process are at the core of flood forecasts. To date operational flood forecasting models use areal rainfall estimates that are based on geographical

features. This paper introduces a new methodology to optimally blend the weighting of gauges for the purpose of obtaining superior flood forecasts. For a selection of 7 Australian catchments this methodology was able to yield improvements of $15.3\%$ and $7.1\%$ in optimization and evaluation periods respectively. Catchments with a low gauge density, or an overwhelming majority of gauges with a low proportion of observations available, are not well suited to this new methodology. Models which close the water balance and demonstrate internal model dynamics that are consistent with a conceptual understanding of the

rainfall-runoff process yielded consistent improvement in streamflow simulation skill.

## 1 Introduction

Rainfall-runoff models form the basis of hydrolgical simulations for a wide range of applictions, extending from forecasting floods through to water resources assessments and the design of water management structures. Hydrological models generate streamflow simulations from forcing data including rainfall. The quality of any hydrological simulation is dependent on the

availability of high quality estimates of areal rainfall for model calibration and simulation.

For rainfall-runoff models the concept of *garbage in, garbage out* is inescapable. An improper treatment of forcing data leads to a cascade of errors which renders all other modeling techniques ineffective (Kuczera et al., 2010). Rainfall is the primary forcing for flood forecasting models and can be derived from in-situ gauges, satellite and ground mounted RADARs, and re-analysis products. The quality of observations, their quality control (QC), and the methods to aggregate rainfall observations

all play a key role in the effective identification of hydrological model parameters (Te Linde et al., 2008) and the assimilation (Vanden-Eijnden et al., 2013) of observed states and outputs. Due to relative accuracy, precision, and the near immediate availability of QC'd observations, methods that utilize gauge data to derive areal rainfall incident upon a catchment are essential for operational flood forecasts (Li et al., 2016) and consequently the focus of this study.

The quality of gauge based areal rainfall estimates hinges upon systematic errors, QC of observations, the design of rain

gauge networks, and methodologies to interpolate areal rainfall from the gauge network. In a review of WMO gauge inter-





comparison studies, Sevruk et al. (2009) stressed the need to correct for systematic error sources such as wind, evaporation, and shielding. Rodda and Dixon (2012) highlight that systematic errors from wind contribute most significantly towards rainfall undercatch while Pollock et al. (2018) demonstrated that a conventional cylinder shaped rain gauge mounted at 0.5 m can underestimate rainfall measured at the ground by a pit gauge by more than 23% on average. QC of gauged data sets is often
performed via comparison with re-analysis (Robertson et al., 2015) or RADAR based products (Qi et al., 2016). Rain gauge networks are used at different temporal and spatial scales for meteorological, climatological, and hydrological purposes. Few rain gauge networks are designed to optimize the ability of hydrologic models to simulate streamflow and flood events (Chacon-Hurtado et al., 2017). Dong et al. (2005) demonstrated that local climatic and geographic conditions play a role in the ability of each gauge to improve hydrologic simulation and that some gauges hinder the successful simulation of streamflow. For their
study catchment Xu et al. (2013) found that hydrological simulation skill improved significantly until a gauge density of 0.4 gauges per km$^2$ was reached and that simulation skill degraded for rain gauge densities less than 1.4 gauges per km$^2$. This would either indicate that systematic errors have not been sufficiently QC'd or that the methodology to interpolate areal rainfall from the gauge network does not adequately weight important gauges. Therefore this study focuses on determining a superior interpolation method that adequately weights the important rainfall gauges.
There are a variety of interpolation methods used to obtain an areal rainfall estimate from a gauged network ranging from simple methods such as the Theissen's polygons and inverse distance weighting, to more complex methods such as kriging. Ly et al. (2013) found no optimal spatial interpolation method for operational hydrology, and that interpolation performance often depended on the density of the gauge network, the spatial and temporal resolution of the data, and parameters such as those used by the semi-variogram in kriging. Mair and Fares (2011); Chen et al. (2017), and Liu et al. (2018) compared interpolation
methods based on their ability to simulate streamflow, with each study using a different hydrological model and catchment. In respective order they found that ordinary kriging (OK), principal component regression with residual correction (PCRR) and inverse distance weighting (IDW) interpolation of gauged rainfall data yielded superior streamflow simulations. However, none of the studies included all three methodologies. It is hypothesized that the best performing interpolation methods are not optimized for hydrological simulations and that their success iss likely to be greatly influenced by the choice of hydrological
model, chosen catchment, and simulation metric.

The search for an optimal interpolation of rainfall gauges for hydrological simulation is essentially an inverse modeling question that asks, how do hydrologists extract optimal rainfall forcings from streamflow simulations? Methods to extract optimal rainfall from streamflow range from those that do not need rainfall forcing such as a cascading linear inversion of runoff (Hino, 1986) to those that require rainfall forcing such as linear inversion (Kirchner, 2009), optimization of storm
multipliers (Kavetski et al., 2006; Vrugt et al., 2008), reduction of forcing data (Wright et al., 2017), and the inclusion of ancillary data (Wright et al., 2018). The nature of these studies imply that rainfall obtained from interpolation methods is not *true* rainfall. However, no example is found in the literature which presents a methodology to optimally interpolate rainfall gauge data for hydrological simulations and provide insights derived therein. To develop an algorithm for the optimization of gauge weights (OGW) that allows the estimation of rainfall which optimizes hydrological simulation skill it is necessary to
include additional degrees of freedom in the parameter optimization process. Many hydrological purists are strong advocates of





*Occam's Razor* and are, rightfully so, concerned that adding hydrological parameters will result in over-fitting of hydrological models. Keeping this in mind the value of additional parameters should be determined through the ability of those parameters to provide improved solutions for hydrological simulations in either a evaluation or forecast period.

Consequently, this paper contributes to the hydrological sciences by developing a novel OGW method which is used in the estimation of areal rainfall for hydrological simulations and providing insights on the potential influences that catchment characteristics and the choice of hydrological model pose for areal rainfall estimates based on gauge interpolation.

## 2    Data set

This study uses seven Australian catchments that are susceptible to flooding, and have different rainfall gauge densities, climatology, area, response times, flow characteristics, and seasonality of rainfall. Monthly PET data from the Australian Water

Availability Project (AWAP) (Raupach et al., 2009, 2012) were used and disaggregated at a uniform hourly rate for each month. This disaggregation strategy is appropriate because streamflow simulations are relatively insensitive to the temporal resolution of PET (Samain and Pauwels, 2013). Both hourly rainfall and streamflow data (Bureau of Meteorology, 2019b) were obtained from the Bureau of Meteorology (BoM) with the catchments being delineated using the Australian geofabric data set (Bureau of Meteorology, 2019a). All rainfall gauges within 10 km of the catchment boundary were considered. A summary of the

catchment properties is given in Table 1, whilst an indication of the available observations for each rain gauge in each catchment is given in Figure 1. Ephemeral streams were defined as those having zero flows in $> 4\%$ of their records (Bennett et al., 2016), and Köppen Gieger climate classifications were taken from Peel et al. (2007). Brief summaries of each catchments main characteristics are as follows:

– *Hurdle Creek at Bobinawarrah* has a small catchment that drains to an ephemeral tributary stream of the frequently
flooded Ovens River in north-eastern Victoria. It has a temperate climate and a near uniform seasonal rainfall distribution.

– *Onkaparinga River upstream of Hahndoorf dissipator* lies near the head of the flood prone Onkaparinga River catchment. The ephemeral stream is located within a temperate climate in south-eastern South Australia (SA) and has winter rains with drier summers.

– *South Esk River at Llewellyn* is part of a river system with a long history of flooding. The intermittent stream is located
in north-eastern Tasmania and belongs to a temperate climate that has a near uniform seasonal rainfall distribution.

– *Clarence River at Paddys Flat* is an intermittent stream at the headwaters of a flashy catchment that observed its worst flood in January of 2013. The catchment lies in the eastern part of New South Wales (NSW) just south of the Queensland border in a temperate climate that has winter rains and drier summers.

– *Tully River at Euramo* is a perennial stream that has been consistently subject to moderate floods over the last two
decades. It lies within a tropical monsoon climate in far north Queensland that has the bulk of its rainfall over summer.





– *Condamine River at Warwick* is an ephemeral stream that contributes to the slow moving Condamine River. The Condamine River has seen some of the countries worst flooding, having had 3 majors floods in the last decade alone. The catchment lies north of the New South Wales (NSW) in the eastern part of Queensland in a temperate climate that has winter rains and drier summers.

– *Isaac River at Yatton* is an ephemeral stream that lies within a hot semi-arid climate in north Queenslad that is subject to large summer rainfall events. The Isaac River is a tributary of the Mackenzie and Fitzroy Rivers and consequently plays a vital role in flood warning. The catchment of Yatton has been subject to 3 major floods in the last decade.

## 3 Methods

In this study, the effects of two different methods of estimating areal rainfall on streamflow simulation performance are com-
pared using three different hydrological models. Here the methods of estimating areal rainfall are introduced, followed by a description of the hydrological modelling approach and techniques for evaluating streamflow simulation performance.

### 3.1 Areal rainfall estimation

In essence the estimation of areal rainfall $r_i$, at time step i, from rainfall gauges involves adequately estimating the weighting of gauges and is determined by

$$r_i = \sum_{j=1}^{n} (r_{i,j}. \times w_{i,j}), \tag{1}$$

where $r_{i,j}$ and $w_{i,j}$ are the rainfall volume and gauge weighting for each timestep i and gauge j respectively. The. $\times$ oeprator indicates element by element multiplication. For each timestep where a gauge has on observation, the gauge weights $w_{i,j}$ are automatically scaled to unity by

$$w_{i,j} = f_{i,j}. \times w_j./(f_{i,j} \times w_j), \tag{2}$$

where $w_j$ is the gauge weight for each location and $f_{i,j}$ is the binary filter array which describes if a gauge has an observation at each time step.

### 3.1.1 Inverse distance weighting

In contrast to the OGW method, both gauge weights and areal rainfall estimation are calculated prior to the estimation of model parameters. Gauge weights were calculated by

$$w_j = 1/d_j^r, \tag{3}$$

where $d_j$ is the distance from gauge j to the catchment centroid and r is a power parameter taken to be 2 for this study.





### 3.1.2   Optimization of gauge weights

Methods such as IDW do not consider the hydrological model performance in estimating areal rainfall. To consider the effects of areal rainfall estimation on streamflow simulations an approach to optimize gauge weights as a part of the hydrological model calibration process is introduced. To allow for the identification of optimal gauge weights via a parameter estimation algorithm, maximum and minimum parameter bounds need to be identified. During the optimization process, if weighting for a gauge were able to become zero then data for that gauge is effectively discarded. If this were able to occur it is possible that the chosen likelihood function becomes maximized by discarding multiple gauges and assigning an unrealistic weight to one gauge. In this situation the hydrological model parameters could be over-fitted. Over-fitting is defined to occur when the addition of parameters leads to improved performance in the optimization periods and decreased performance in the evaluation periods. To avoid this gauge weights, $w_j^{min}$, were assigned a minimum un-scaled value of one. It is then logical that gauges which have observations available for a larger proportion of time to potentially have larger gauge weightings. To ensure that the gauges with less observations contribute less to areal rainfall estimates, this maximum un-scaled gauge weighting reduced according to an appropriate negative exponential. The maximum allowable un-scaled gauge weightings $w_j^{max}$ that could be explored were calculated as

$$w_j^{max} = \max(1.01, (n-1) \times 0.55/0.45 \times (1 - 2.03 \times e^{-4.74 \times p_j})), \tag{4}$$

where $n$ is the total number of gauges used and $p_j$ is the proportion of observations available for gauge $j$. Equation 4 can be modified to allow different parameter spaces be searched for each gauge, that has a higher or lower proportion of rainfall observations. Lastly, $w_j^{max}$ were restricted to values that ensured a maximum scaled weighting of $0.55$ if that gauge had all observations available.

### 3.2   Hydrological models

The hydrological models used in this study were chosen based on their proven performance, different routing schemes, and complexity. Both the modèle de Génie Rural à 4 paramètres Horaire (GR4H) and Hydroliska Byråns Vattenbalansavdelning (HBV) model use unit hydrograph (UH) routing whilst the Probability Distributed Model (PDM) uses a dynamic storage based routing approach. Both HBV and PDM close the water balance whilst GR4H does not (Perrin et al., 2003).

### 3.2.1   GR4H

The GR4H is an hourly application of the modèle de Génie Rural à 4 paramètres Journalier (GR4J) model (Perrin et al., 2003) which utilizes two state variables, soil water storage and routing water storage. The model is mathematically parsimonious and consists of 4 parameters which, along with the state variables, describe evapotranspiration, percolation, and both slow and fast runoff. A potential pitfall of the model is that the catchment exchange term allows for either the import or export of water into the model. Thus the water balance is not closed and potential biases in observation data corrected for.





### 3.2.2 PDM

The PDM as described by Moore (2007) consists of a set of functions used to describe various hydrological systems. The model consists of four states, a cascade of two linear stores which are used to describe surface runoff, a linear store used to 150 describe subsurface flow and a catchment based soil moisture store that is considered to consist of soil moisture stores with varying capacities that are able to be represented by a Pareto distribution. The PDM uses 9 model parameters and routing is a dynamic storage based process.

### 3.2.3 HBV

Of the many HBV model (Lindström et al., 1997) variants, this paper uses the version developed by Matgen et al. (2006) 155 and used by Pauwels and De Lannoy (2015). The model consists of three state variables, a soil reservoir, a slow reservoir, and a fast reservoir. The 11 parameters, 3 state variables, and subsequent governing equations are used to calculate the actual evapotranspiration, infiltration, effective rainfall, percolation, and proportion of effective rainfall that enters the fast and slow reservoirs, outflow from each reservoir, and UH based routing.

### 3.3 Parameter estimation

The estimation of model and OGW parameters was performed simultaneously using the Differential Evolution Adaptive Metropolis algorithm with past state sampling and snooker updating (DREAM$_{ZS}$) of Vrugt (2016). The default settings outlined in Vrugt (2016) were used. DREAM$_{ZS}$ finds the posterior parameter distribution which maximizes a chosen likelihood function.

### 3.3.1 Likelihood function

The optimal weighting of rainfall gauges is specific to the chosen likelihood function and will change if a likelihood function which places priority to low flows is chosen instead of a likelihood function that places priority on high flows. The same can be said if objective functions are maximized instead of likelihood functions. Since this paper is primarily concerned with the OGW for flood events, a Gaussian likelihood function (Thiemann et al., 2001) is used.

### 3.4 Evaluation strategy

The evaluation of optimized parameters and gauge weightings was carried out via split sample testing. A warm-up period of one year was used. The skill of streamflow simulations were then evaluated for the six year period beginning 1$^{st}$ of January 2008 and ending 31$^{st}$ of December 2013. The optimization and evaluation periods were 5 and 1 years long respectively. The optimization and evaluation periods were alternated until all 6 years of evaluation data had been used for evaluation. This resulted in a total of 6 split samples for each catchment.





Model results are presented and evaluated using the Root Mean Square Error (RMSE) metric. RMSE is given by

$$\text{RMSE} = \sqrt{\frac{\sum_{i=1}^{t}(q_i^s - q_i^o)^2}{t}}, \tag{5}$$

for which t represents the length of the optimized simulation period, and $q_i^s$, and $q_i^o$, represent the simulated, and observed streamflow at time step i respectively. Relative improvements in simulation skill are determined by calculating the ratio of RMSE between the stated cases.

## 4    Results and discussion


The results of the split sample testing are presented as an RMSE average of the samples for each catchment and model for the optimization and evaluation periods in Tables 2 and 3 respectively. The HBV model consistently yields lower RMSE values for the optimization period using both the IDW and OGW rainfall products. As expected, the addition of extra parameters for the optimized rainfall product consistently leads to notable improvements throughout the optimization period. There are however a few split samples in which the additional parameters used in the OGW methodology led to a marginal increase in RMSE. This can happen when the scaling of gauge weightings does not allow the optimization process to attain the same gauge weights as the IDW method.


Whilst the OGW rainfall estimates generally led to improvements in the evaluation period, there was much less consistency in performance between models. For IDW rainfall the GR4H model had the lowest RMSE for 5 out of the 7 catchments. The best simulations were obtained when OGW rainfall was used for 5 out of the 7 catchments. On average the observed relative improvements in the evaluation period were $-7.9\%$, $6.2\%$, and $8.3\%$ for the GR4H, HBV, and PDM models, respectively. If the best IDW simulation is compared to the best OGW simulation an average improvement of $4.3\%$ was observed.


To uncover the reasons for the apparent mismatch in performance between the optimization and evaluation periods it is necessary to reflect on the methodology and available data. The two catchments in which the IDW simulations performed better than the OGW simulations both have deficiencies in data availability. Table 1 and Figure 1 show that the Onkaparinga catchment has 21 gauges, only one of which has data available for more than $50\%$ of the time. Since they are more able to contribute to an increase in simulation skill in the optimization period, gauges with a larger proportion of observations available in the optimization period are more likely to obtain higher weights. If these gauges have an overall low proportion of observations available they are more likely to not have observations available in the evaluation period. This is likely to lead to an increase in simulation skill for the optimization period but not for the evaluation period. This improvement in skill in the optimization period and decrease in skill in the evaluation period was observed in the Onkaparinga catchment, a catchment with the overwhelming majority of gauges having $< 50\%$ data availability. Consequently this technique is not recommended for catchments that have a overwhelming majority of rainfall gauges with poor ($< 50\%$) data availability. Aside from the Onkaparinga catchment the catchment of Yatton is the only other catchment in which the IDW rainfall simulations perform better than the OGW simulations in evaluation periods. The Yatton catchment has one gauge per 2806 km$^2$ whereas every other catchment has at least one gauge per 200 km$^2$. Zeng et al. (2018) demonstrated that identification of model parameters became








increasingly difficult as the density of the rain gauge network decreased. This effect is made worse by the OGWs. Consequently, a poor gauge density does not allow for adequate estimation of hydrological model parameters, gauge weightings, and areal rainfall. If the results of these two catchments are not considered, comparing the best OGW simulation to the best IDW simulation leads to improvements of 15.3% and 7.1% in the optimization and evaluation periods, respectively.

Applying all three models and selecting the best performing model resulted in the OGW leading to considerable improvements. It is however concerning to see that the simulation skill in the evaluation periods consistently decreaseed for the GR4H model. If some models are not well suited to be used with OGW rainfall estimates then it is imperative to know which elements of the model structure contribute to this decrease in skill. The average change in rainfall and net forcing, where net forcing are defined as rainfall minus evapotranspiration, between IDW and OGW simulations, are shown in Figure 2 and Table 4 respectively. OGW rainfall is larger than IDW rainfall for all hydrological models and catchments, except Yatton. It was also common for both HBV and PDM to obtain larger OGW rainfall estimates than GR4H. This trend is in stark contrast to the pattern in the change of net forcing. The mean and standard deviation of the change in net forcing across catchments was significantly lower for HBV and PDM than it was for GR4H, indicating that the internal dynamics of the HBV and PDM adapt to a change in rainfall by altering evapotranspiration while GR4H does not.

## 4.1 Case Study - Paddys Flat

This case study was designed to develop a deeper understanding of the impact the OGW and subsequent areal rainfall estimates had on the hydrologic model simulations and internal dynamics. Results and their associated discussion are presented sequentially to describe the impacts on rainfall, model parameters, the water balance, and streamflow. The observed streamflow and IDW areal rainfall estimate for Paddys Flat can be seen in Figure 3.

The comparison of the cumulative rainfall volumes obtained for the split sample simulations for each model at Paddys Flat is shown in Figure 4 demonstrating that the OGW leads to a consistent increase in cumulative rainfall volumes. Each split sample obtained through model evaluation is represented by a different line. The cumulative rainfall volumes estimated by GR4H are less than those estimated by HBV and PDM. However, it is necessary to determine if there is a tendency for low or high magnitude rainfall observations to be estimated differently by the OGW or IDW interpolation methods.

Plots of the OGW estimation of areal rainfall obtained by each model versus the IDW estimation of areal rainfall in Figure 5 reveal that, when compared to the IDW interpolation method, the OGW methodology did not have a tendency to predict greater rainfall volumes for small or large magnitude rainfall observations. The larger cumulative rainfall volumes observed by the OGW method were therefore a result of a slight tendency to estimate greater rainfall volumes for both small and large magnitude rainfall observations. The impact each gauge has on the estimation of areal rainfall was then explored.

Hydrological simulation skill based OGWs allows gauges which add value to forecasts to be identified. The gauge weightings determined by the IDW interpolation method and OGW method for each split sample and model are shown in Figure 6. Each model selected similar gauges and weightings for the majority of gauges. There was however one or two gauges which only one model assigned weight to. Therefore the results from the analysis of rainfall estimates obtained through the OGW method do not indicate that GR4H, HBV, and PDM require model specific rainfall forcings, and nor do they give sufficient reasoning





as to why the OGW simulations decrease in simulation skill for GR4H and improve in simulation skill for HBV and PDM. It is therefore likely that some difference in model parameters and subsequent impact on internal model dynamics caused the increase/decrease in skill.

To demonstrate the change or lack of change in model parameters that were observed for each rainfall estimation method, an average of the simulated split sample model parameters for the OGW and IDW simulations for Paddys Flat is shown in Figure 7. GR4H parameters did not noticeably alter whilst parameters that influence soil moisture content, evapotranspiration, and percolation change significantly for both HBV and PDM. Parameters influencing fast flow and baseflow change for HBV and PDM respectively. The change in these specific parameters indicate that internal model dynamics are sensitive to changes in forcing data for both HBV and PDM but not for GR4H. This presents a basis as to why rainfall minus evapotranspiration change much less for both HBV and PDM than GR4H.

The water balance is given as

$$s_{\mathrm{i}} = \sum_{m=1}^{i} r_{\mathrm{m}} - q_{\mathrm{m}} - e_{\mathrm{m}}, \tag{6}$$

where $s_{\mathrm{i}}$ and $e_{\mathrm{m}}$ are the catchment storage, and evapotranspiration at time steps i and m respectively. It should be noted that for GR4H a catchment exchange process that either abstracts or provides additional water may occur. Figure 8 shows the water balance for the split sample simulations for the OGW and IDW methods for each model at Paddys Flat. After the warm-up period of one year, simulations for both the HBV and PDM models reached a point of equilibrium in which a rise or decline in $s_{\mathrm{i}}$ was eventually countered by a subsequent and opposing rise or decline in $s_{\mathrm{i}}$. This trend is the cornerstone of a model that is able to close the water balance. A non-zero positive or negative catchment exchange parameter in GR4H leads to simulations which do not close the water balance. This is the reason why $s_{\mathrm{i}}$ gradually increased for GR4H. Referring back to Figure 7 it can be seen that the catchment exchange parameter was marginally above the mean of the minimum and parameter bounds. This mean is zero and explains why there is only a gradual departure from zero.

The IDW storage profiles and the OGW storage profiles obtained using GR4H remained inconsistent with the current conceptual understanding of the rainfall-runoff process. The storage profiles for GR4H have different trajectories for each split sample which became more dispersed with the inclusion of OGW rainfall estimates. In contrast, both HBV and PDM demonstrated storage profiles that tend towards an equilibrium. The storage profiles of HBV and PDM for each split sample are more similar to each other than when the IDW rainfall estimates were used. Further, the storage profiles obtained using the HBV and PDM were remarkably similar for all split samples regardless of whether or not the IDW or OGW rainfall estimates were used. As such both, HBV and PDM demonstrate improved internal model dynamics which are consistent with a conceptual understanding of the rainfall-runoff process. Lastly, the ability of each model to represent different streamflow events was analyzed.

The OGW estimation of areal rainfall enabled both HBV and PDM to simulate internal dynamics with improved consistency. Models that have more consistency in their representation of internal dynamics are better positioned to benefit from the inclusion of soil moisture data for calibration and/or assimilation purposes. Similar benefits are likely to be observed when





updating internal states through the assimilation of observed streamflow. The relative impact of the OGW methodology on
275  streamflow, when compared to the IDW methodology, can be observed in Figure 9. It is clear that for larger magnitude events
the OGW methodology brought the streamflow simulations closer to observations for the HBV and PDM throughout the eval-
uation period. Conversely, the use of OGW rainfall areal rainfall estimates with GR4H led to marginally improved streamflow
simulations for one event and significantly worse streamflow simulations for another event. This result was observed at all
catchments with sufficient gauge density and proportion of observations available.

280  **5   Conclusions**

This study developed a methodology to optimally weigh rainfall gauges such that improved flood forecasting skill could be
obtained for three different conceptual hydrological models. The OGW methodology developed was tested on seven Aus-
tralian catchments and a comparison of streamflow simulations obtained using the IDW and OGW methods demonstrated an
improvement in streamflow RMSE of $15.3\%$ and $7.1\%$ in the optimization and evaluation periods respectively, for catchments
285  that have multiple rainfall gauges with observations available $> 50\%$ of the time and more than one rainfall gauge every 200
$km^2$. The methodology did not work equally well for the three hydrological models chosen. Improvements in evaluation peri-
ods were only noticed for the PDM and HBV hydrological models, and not for the GR4H hydrological model. The most likely
explanation for this is the inability of the GR4H model to represent internal dynamics that are consistent with a conceptual un-
derstanding of the rainfall-runoff process. This methodology opens new possibilities for model evaluation, forcing uncertainty
290  and data assimilation studies in hydrology.

*Author contributions.*  AW conducted the experimental work, contributed towards the theory and wrote the paper. JW assisted in the writing
process. VP and DR contributed towards the theory and assisted in the writing process.

*Acknowledgements.*  The authors would like to extend their gratitude to the anonymous reviewers for their comments and recommendations.
This work was supported by the Multi-modal Australian Sciences Imaging and Visualisation Environment (MASSIVE) (http://www.massive.org.au).





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





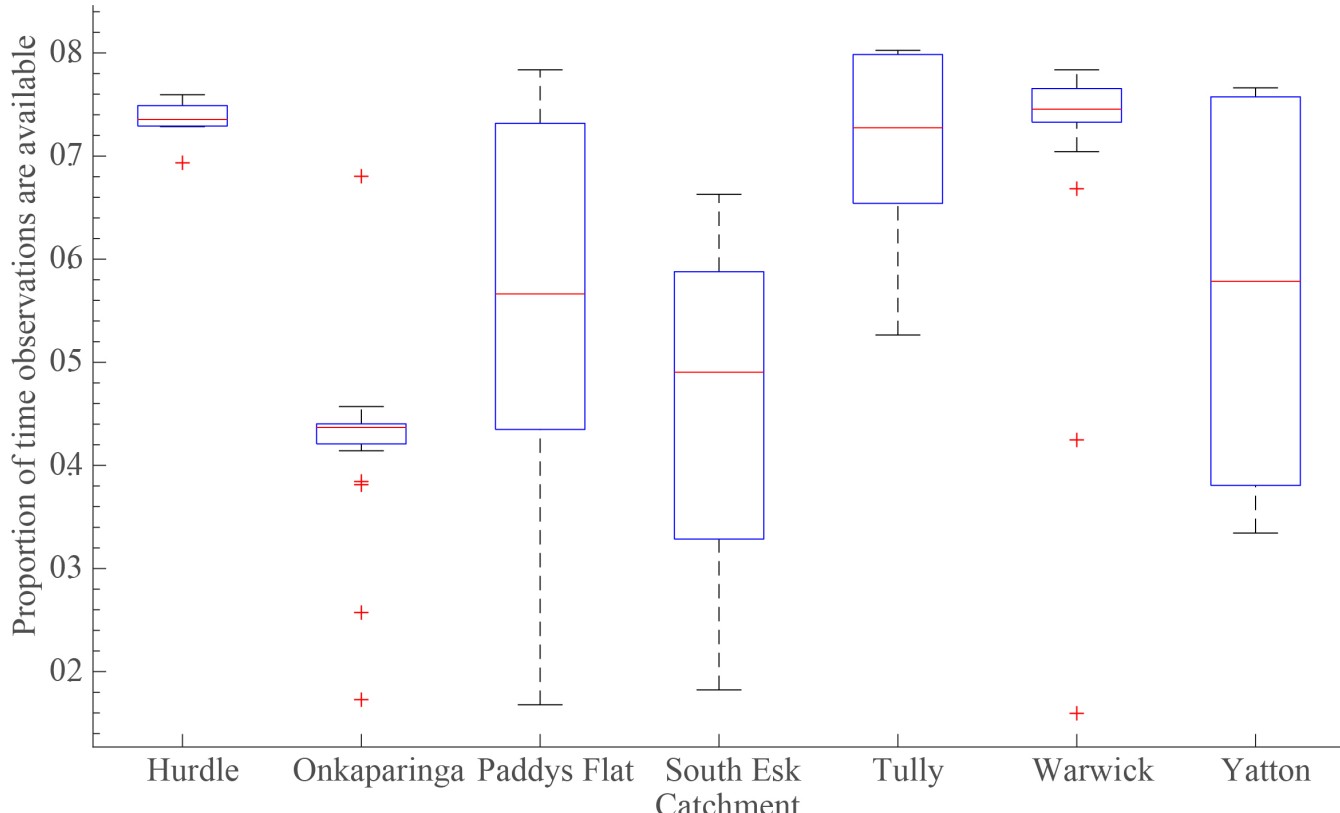

**Figure 1.** Box plots showing the median, $25^{th}$ and $75^{th}$ percentile, end points and outliers that describe the proportion of observations available for gauges within each catchment.

**Table 1.** Catchment area, number of rainfall gauges, and the % of time that no gauge has an observation for each of the study catchments.

| Catchment | Number of gauges | % of time no gauge works | Catchment area km$^2$ |
|---|---|---|---|
| Hurdle Creek | 12 | 15.2 | 156 |
| Onkaparinga | 21 | 22.5 | 208 |
| South Esk | 13 | 17.5 | 2289 |
| Paddys Flat | 27 | 2.7 | 3112 |
| Tully | 7 | 4.1 | 1390 |
| Warwick | 21 | 5.3 | 1379 |
| Yatton | 7 | 3.6 | 19639 |





**Figure 2.** The average change in estimated areal rainfall for the simulation period between the OGW and IDW areal rainfall estimation methods.

**Table 2.** Average RMSE [mm] for simulated streamflow in the optimization periods for each catchment using IDW and OGW rainfall. Values in bold indicate the lowest obtained RMSE.

| Catchment | GR4H | | HBV | | PDM | |
| --- | --- | --- | --- | --- | --- | --- |
| | IDW | Opt | IDW | Opt | IDW | Opt |
| Hurdle Creek | 0.033 | 0.031 | 0.032 | **0.028** | 0.033 | 0.028 |
| Onkaparinga | 0.040 | 0.039 | 0.037 | **0.037** | 0.039 | 0.039 |
| South Esk | 0.037 | 0.033 | 0.035 | **0.032** | 0.043 | 0.039 |
| Paddys Flat | 0.068 | 0.053 | 0.061 | **0.048** | 0.084 | 0.069 |
| Tully | 0.175 | 0.177 | 0.165 | **0.151** | 0.170 | 0.157 |
| Warwick | 0.032 | 0.023 | 0.032 | **0.023** | 0.047 | 0.038 |
| Yatton | 0.062 | 0.056 | 0.053 | **0.048** | 0.061 | 0.057 |

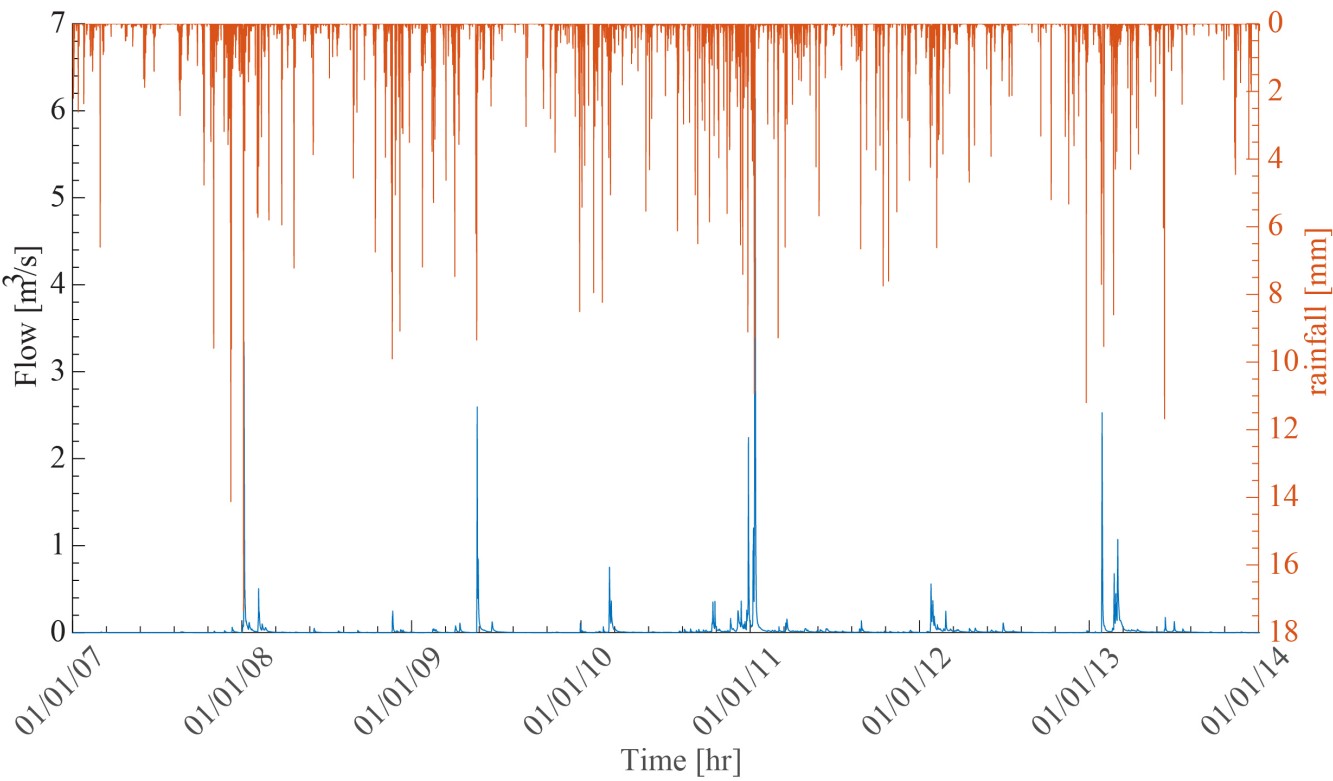

**Figure 3.** Observed streamflow and IDW areal rainfall estimates for Paddys Flat.

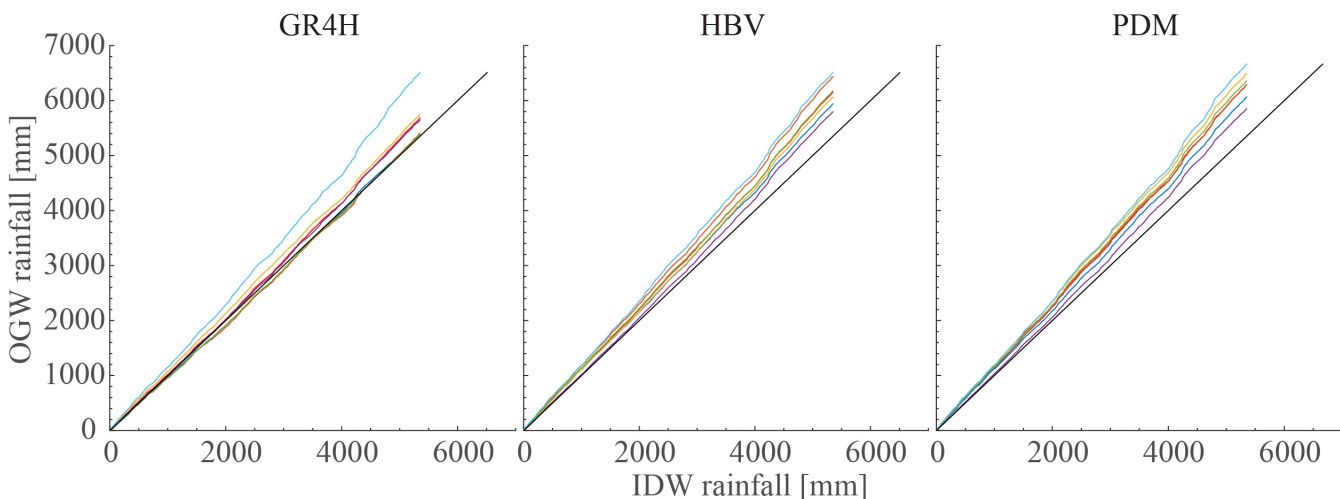

**Figure 4.** Cumulative areal rainfall estimates obtained for each OGW simulation when compared to the cumulative IDW rainfall areal rainfall estimate. 1-1 reference lines are shown in black. Each colour represents a different split sample.

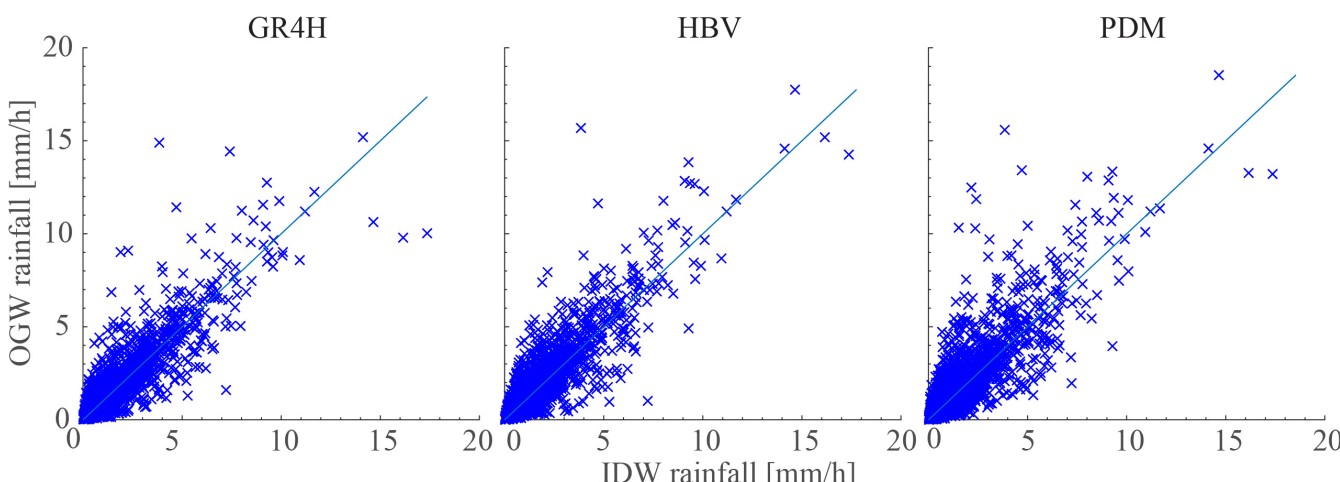

**Figure 5.** A 1-1 comparison of the average OGW areal rainfall estimates with the IDW areal rainfall estimates for each model.

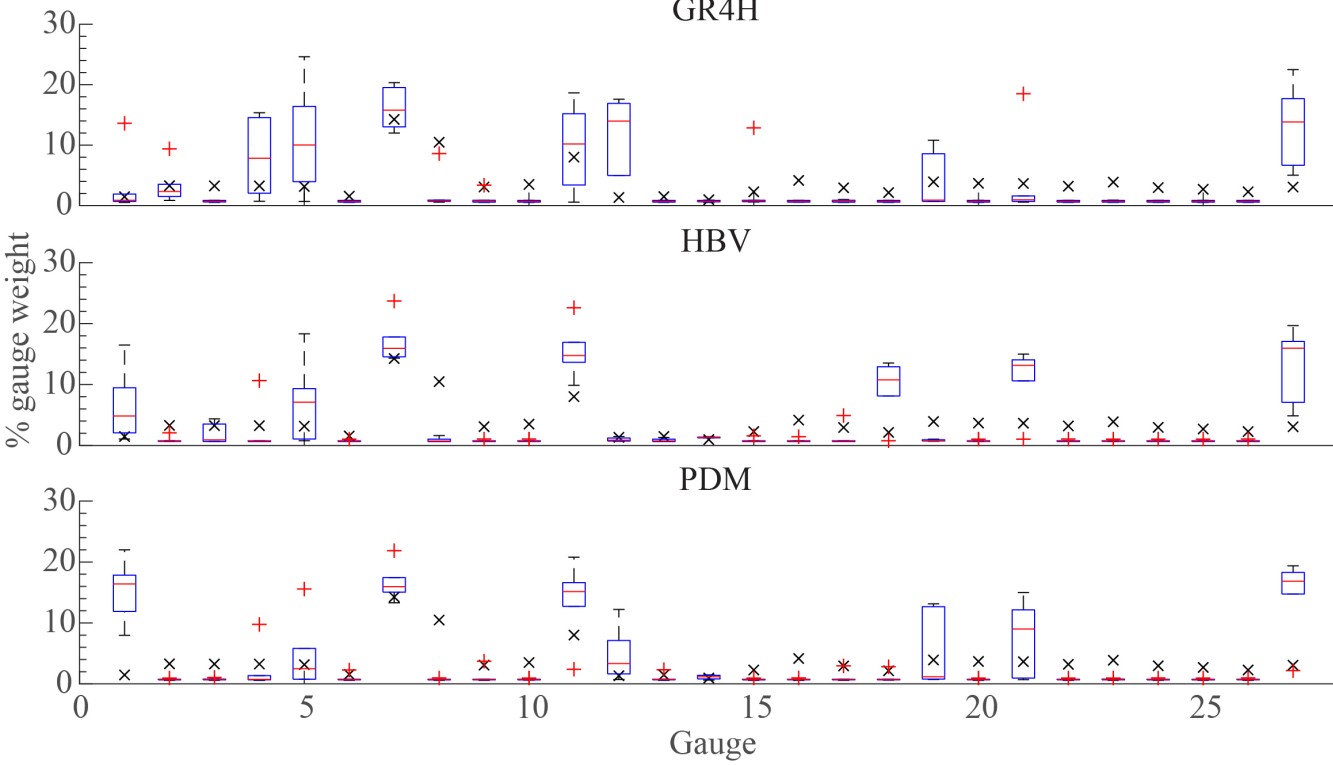

**Figure 6.** A comparison of the IDW gauge weights, as indicated by a ×, with OGW gauge weights for each split sample simulation. A +
indicates a potential outlier.





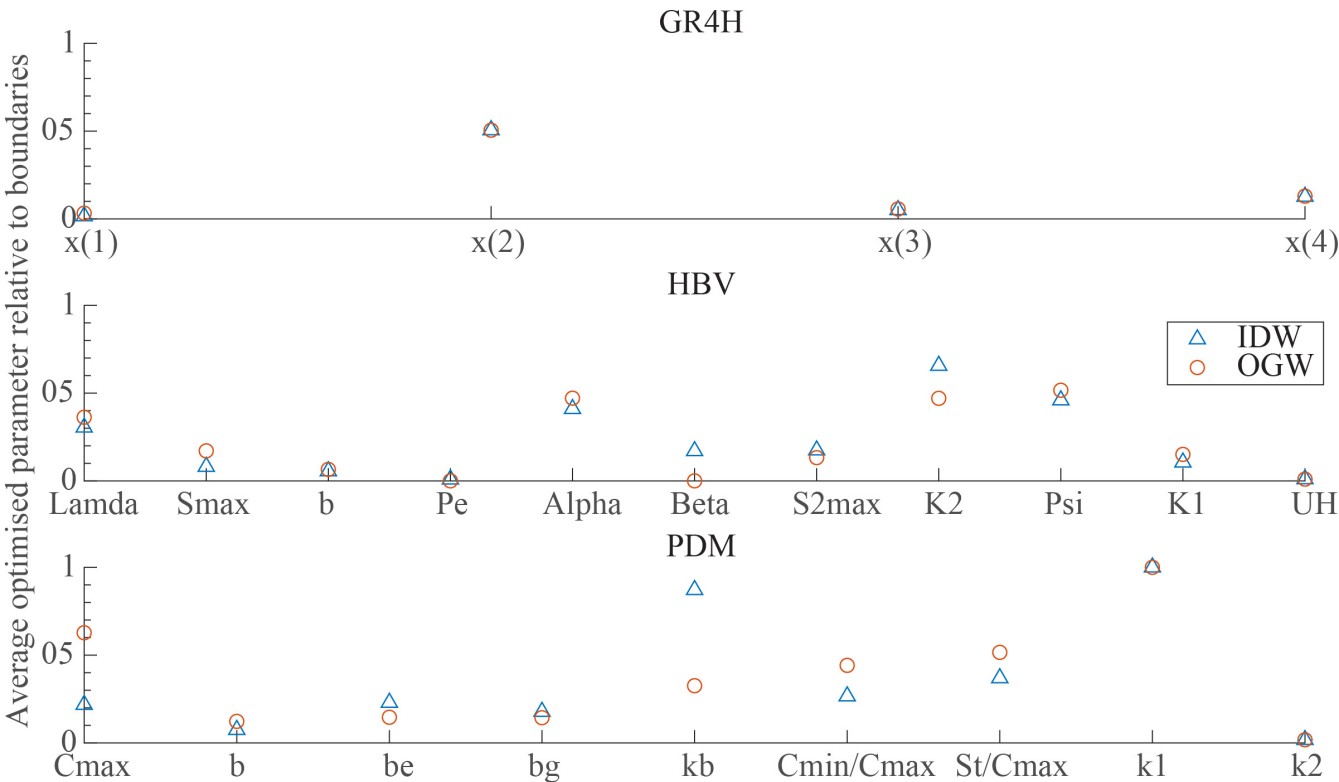

**Figure 7.** Relative average optimized parameters for each rainfall-runoff model using OGW and IDW rainfall estimates. Values of 0 and 1 indicate the lower and upper bounds for each parameter respectively.

**Table 3.** Average RMSE [mm] for simulated streamflow in evaluation periods for each catchment using IDW and OGW rainfall. Values in bold indicate the lowest obtained RMSE.

| Catchment | GR4H | | HBV | | PDM | |
|---|---|---|---|---|---|---|
| | IDW | Opt | IDW | Opt | IDW | Opt |
| Hurdle Creek | 0.033 | 0.033 | 0.041 | 0.036 | 0.037 | **0.029** |
| Onkaparinga | **0.035** | 0.036 | 0.038 | 0.039 | 0.038 | 0.039 |
| South Esk | 0.037 | **0.037** | 0.044 | 0.043 | 0.053 | 0.044 |
| Paddys Flat | 0.083 | 0.103 | 0.082 | **0.077** | 0.093 | 0.080 |
| Tully | 0.176 | 0.184 | 0.161 | **0.151** | 0.167 | 0.156 |
| Warwick | 0.035 | 0.038 | 0.042 | **0.031** | 0.050 | 0.041 |
| Yatton | **0.068** | 0.075 | 0.069 | 0.072 | 0.068 | 0.078 |





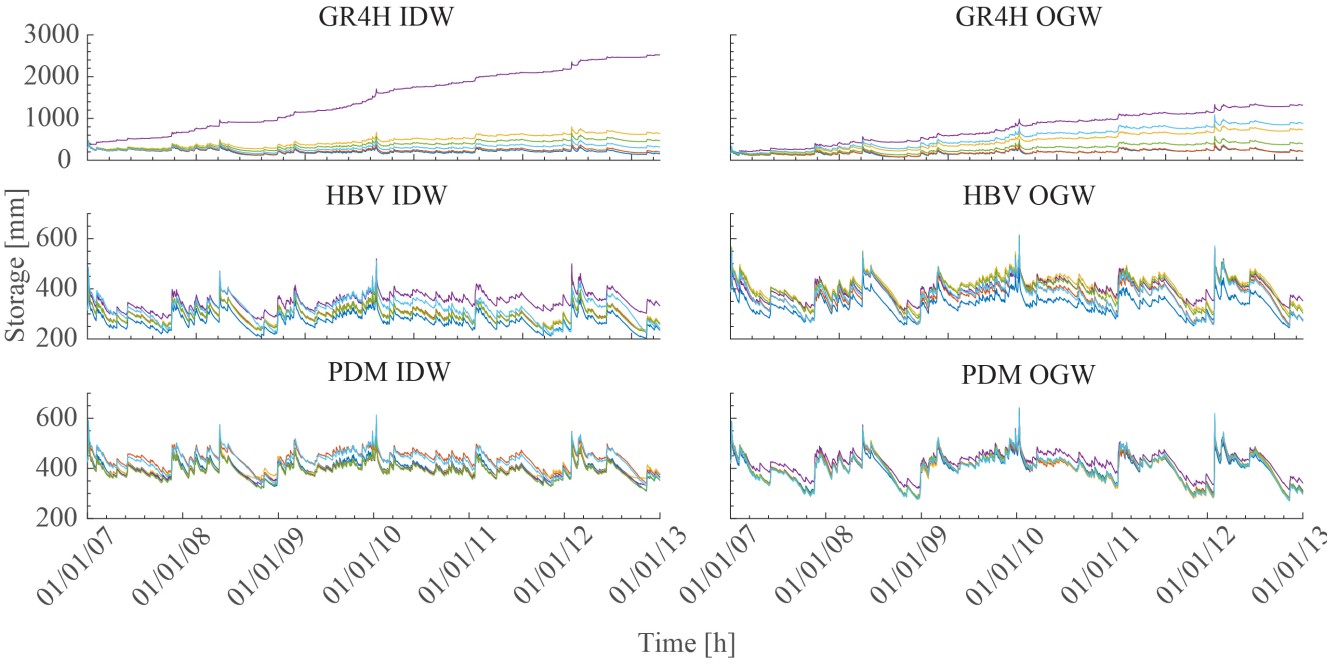

**Figure 8.** A representation of each models ability to resolve storage for each split sample and areal rainfall estimation method respectively. Each time series starts at the end of the warmup period and is adjusted to have the same storage as other samples from the model and rainfall estimation method pair. Each colour represents a different split sample.

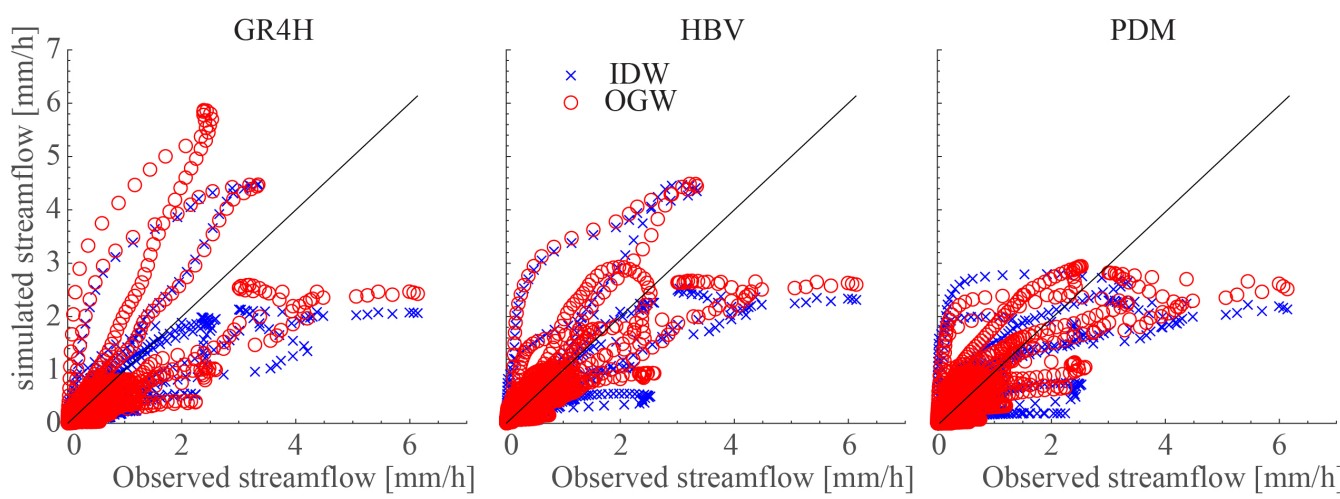

**Figure 9.** A comparison of the streamflow simulations for the evaluation periods for each model and areal rainfall estimation method. 1-1 reference lines are shown in black.



**Table 4.** Average mm change in rainfall minus evapotranspiration between the IDW and OGW rainfall simulations. A positive change indicates that net model forcings were higher for the OGW simulations. StDev represents the standard deviation.

|  | GR4H | HBV | PDM |
|---|---|---|---|
| Hurdle Creek | 319.846 | 160.691 | 138.951 |
| Onkaparinga | 37.993 | 3.988 | 44.639 |
| South Esk | 383.170 | -40.262 | 107.997 |
| Paddys Flat | -91.241 | 24.374 | -232.839 |
| Tully | 5524.877 | 1412.470 | 1324.449 |
| Warwick | 65.607 | 18.264 | 97.857 |
| Yatton | 808.369 | -80.898 | 157.520 |
| Mean | 1006.946 | 214.090 | 234.082 |
| StDev | 1864.838 | 494.132 | 461.700 |