# Peer review of "Insights from a new methodology to optimize rain gauge weighting for rainfall-runoff models"

_Hydrology and Earth System Sciences, 2019_

## Referee Comment (RC1) · Anonymous Referee #1 · 15 Oct 2019

The study introduces a new methodology to optimise gauge weights as a part of hydrologic model calibration processes. The study also shows that areal rainfall estimation obtained from the new optimisation led to improved performance of rainfall-runoff models, compared to the traditional way (i.e., IDW). This is an interesting approach to produce optimised rainfall forcing for better hydrologic modeling. However, there are some places in the manuscript that need further clarification.

Major comments:

Extrapolation capacity of model parameters

The new optimisation methodology suggested in the study decides gauge weightings based on current gauge configuration and the proportion of available observations (eq.

r version

4), and the optimisation method affects model parameter values (as shown in Figure 7). In this context, can we guarantee that the model parameters perform well also with areal rainfall estimated from other rainfall data sources (e.g., reanalysis for future projection), compared to the model parameters obtained from the IDW? Moreover, when the gauge configuration is changed (e.g., more gauges are installed in the future), how can we use the obtained model parameters & gauge weightings? Do we have to calibrate models again with the new gauge set-up? Can you address this point with more details?

Specific comments:

p.3 l.69: why PET is considered here? Is it one of forcing data for models? Please clarify.

p.3/Table3: can you specify what time period you considered for each catchment? i.e., the length of 100% time used to calculate % of time that no gauge has an observation.

p.5 eq.4: how do other constant values are decided? Do they have any (physical) meaning for the calculation?

p.7 l.175: Why don't you consider more performance metrics (e.g., correlation, bias of annual runoff) to evaluate the performance of IDW vs OGW from various aspects?

p.8 l.219-220: this point should be more carefully addressed. This also could mean that the model parameters are forced, to some extent, to describe runoff better, regardless of other hydrologic variables (i.e., over-fitting to runoff).

Figure 1. How did you deal with the time steps with missing observation, especially for the Onkaparinga catchment where most of gauges have less than 50% of time series during the study period, so probably there are some time steps when all gauges are not available?

Figure 2 vs Figure 4: given that HBV and PDM shows very similar results in Figure 4 (i.e., OGW increased rainfall), I would expect similar results between the HBV and

PDM also from Figure 2; do you have any idea why both figures are not consistent?

Figure 7: is it showing the average of parameters obtained from the six split samples? Do you also check the "spread" of parameters (e.g., interquartile range) between IDW and OGW?

Minor comments

p.3: you may want to consider providing the map of study catchments

p.2 l.49: iss -> is

p.3 l.69: please explain "PET"

p.4 l.107: on observation -> an observation

p.9 l.260: which one is the catchment exchange parameter in Figure 7?

Figure 1: please make the y-axis label clear, e.g., 0.2 and 0.3 look like 02 and 03, respectively.

Figure 2: can you make it clear that here the change mean "OGW-IDW"?

Figure 4: this is for the Paddys Flat, right? Please clarify.

Figure 8: its't the X-axis from Jan 2008 to Dec 2013?

---

## Referee Comment (RC2) · Anonymous Referee #2 · 17 Oct 2019

**Review of the manuscript HESS-2019-450**

**Insights from a new methodology to optimize rain gauge weighting for rainfall-runoff models**

by
Ashley J. Wright, David E. Robertson, Jeffrey P. Walker, and Valentijn R.N. Pauwels

The authors introduce the OGW (optimization of gauge weights) method and compare their results with the IDW (inverse distance weighting) method, based on data from seven catchments in Australia. Furthermore, results of a case study are presented.

This is an interesting study, which deserves to be published – but the presentation quality should be improved.

I have just noticed that quite some of my potential comments have already been stated by Reviewer # 1, so I repeat just those, which I find most urgent.

**General comments:**

(1) I found the usage units for "streamflow" quite confusing: In Fig.3 it is $m^3/s$, in Fig. 9 it is mm/h. Is this obtained by dividing the flow in [$m^3/s$] through the catchment area? Which units should be used in Equation 5 (for $q$ and $t$)? There is a square root of [streamflow squared, divided by time]. The result can be hardly [mm], as given in Table 2.

(2) Please check the references throughout: The "scopus" information is hardly readable, like: "https://www.scopus.com/inward/record.uri?eid=2-s2.0-84884921481{&}doi=10.1016{%}2Fj.jhydrol.2013.09.004{&}partnerID=

40{&}md5=3554bc7f56537b1b95a3a51160ebe40a, 2013."
I would regards the DOI information as sufficient.

**Specific comments:**

(1) Abstract: "For a selection of 7 Australian catchments this methodology was able to yield improvements of 15.3% and 7.1% .." This is a bit misleading, since you have excluded 2 catchments to get these numbers – right?

(2) Page 2, Line 35: ".. gauge density of 0.4 gauges per $km^2$". Please check the numbers. According to Xu et al. (2013) this should be "per **1000** $km^2$"

(3) P 3, L 68: How did you select the catchments? I understand that all are susceptible to flooding, but are these the catchments, which are most affected?

(4) A map with the locations of the catchments would be very helpful.

(5) You should provide some information on average precipitation in the different catchments. This would help to interpret the results in Fig. 2. Values for Tully look intimidating, but I have just found out that Tully (town) is one of the wettest places in Australia.

(6) P 4, L 116: "taken to be 2 for this study" – why?

(7) P 6, L 158: Please explain "UH"

(8) Figure 1: Values for Hurdle and Warwick seem to be concentrated around ~ 75 %. Do you have an explanation for that? In general I would have expected at least some values closer to 100 %.

(9) Table 4: You give 3 digits after the decimal point, are those digits significant? With such an uneven distribution, the standard deviation (which is much larger than the mean) is not very meaningful.

(10) Table 4: The GR4H values for Tully is 5525 mm – can this be true?

(11) Equation 6: Shouldn't there be a parenthesis? So you would just be summing up the r values – and the symbol r needs to be explained.

(12) Figure 8: One split sample for GR4H IDW shows a totally different behavior. Do you know why?

(13) Figure 4: One split sample for GR4H IDW shows also a different behavior. Do you know why? And is it maybe the same as in Fig. 8?

(14) Figure 9: The PDM model seems to saturate at ~ 2.5 mm/h. Do you have an explanation for this behavior?

**(Some) technical corrections:**

I did not specifically search for typos, but found quite some – so there are probably more.

Some examples:

P 1 L 1: hydrolgical

Same line: applictions

P 2 L 55: The nature …. imply

P 3 L 77: Köppen-Gieger (should be "-Geiger").

P 4, L 95: Queenslad

P 4, L 106: oeprator

P 4, L 127: "to potentially have"    should be "do …" right?

P 5, L 146: "Thus the water balance is not closed and potential biases in observation data corrected for."  seems to be incomplete.

P 6, L 171: "The skill … were .."

P 10, L 281: "weigh" should be "weight" – right?

---

## Referee Comment (RC3) · Anonymous Referee #3 · 22 Oct 2019

This manuscript presents a new method to weigh available rainfall gauges in a way that maximizes the performance of hydrological model in terms of streamflow simulation. The method is tested for 3 hydrological models for streamflow simulation of 7 Australian catchments and its performance is compared to a classic inverse distance weighing (IDW) approach.

The proposed method has its merits and was reported to outperform IDW method in the majority of tested cases. However, the proposed method was compared to only one arbitrary selected interpolation method using only one arbitrary selected performance measure of model efficiency. Little discussion was provided on possible model parameter uncertainty and its effect on the performance of the proposed method. Moreover, the Abstract and Conclusion of the manuscript highlight the importance of the proposed

method for flood forecast, whereas in the performed analysis the emphasis is on the simulation of the historical streamflow time series. Finally, the Introduction lacks presentation of existing methods on rainfall optimization and correction techniques for hydrological modeling. Therefore, I think substantial revision of the manuscript is required before it can be accepted for publication. Below I present my detailed comments.

General comments

1. According to the Abstract and the Conclusion of the manuscript the purpose of the proposed method is to obtain superior flood forecast. However, in the technical part of the analysis the proposed method is not tested for such purposes, but rather for the simulation of historical streamflow time series. In the description of the catchments, noticeable flood events are described in great detail, but none of the information regarding these flood events is used to test the proposed method. The proposed method can be indeed useful for flood forecasting, but since this was not the scope of the analysis I would suggest authors to modify abstract and conclusions accordingly by reporting the results that are actually the outcome of this study.

2. The focus of Introduction is largely on issues regarding systematic errors in rainfall measurement and quality control of these measurements. This is an important topic indeed, but in my opinion current Introduction is disconnected from the proposed method and the performed analysis. Additional information about existing methods on optimization/correction of rainfall would be more welcome (e.g., optimization of areal rainfall (e.g., Anctil et al., 2006) or using precipitation correction factor as a calibration parameter of hydrological models that is often used in data scarce conditions (e.g., Schaefli et al., 2007; Duethmann et al., 2013)).

3. In the Introduction (Line 44-50) the authors show that existing studies do not give a univocal answer on the question which interpolation method result in better streamflow simulation, since the performance of different interpolation methods for streamflow simulation might vary largely when different models, different performance metrics and

different catchment are used. Moreover, the authors highlight that none of the existing studies used all interpolation methods making it difficult to identify the most appropriate method for streamflow simulation. I fully agree with these statements of the authors, but I am surprised that for their own study the authors provided little motivation on the choice of a single baseline interpolation method (i.e., IDW) and a single performance measure (i.e., RMSE) for model evaluation. Moreover, the detailed results were presented only for one out of seven catchments. The differences in performance for different catchments were not discussed in the context of different catchment characteristics and climatic settings.

4. Little attention was paid to the uncertainties that might arise due to the parameter equifinality when the different precipitation inputs are evaluated by goodness-of-fit of simulated streamflow. This is an important topic in the field of precipitation benchmarking (e.g., Yilmaz et al., 2005; Heistermann and Kneis, 2011). The proposed method is likely to suffer from similar problems. Since Figure 7 only report the average value of calibrated parameters for 6 different spilt samples, it is not possible to say if the proposed method is affected by parametric uncertainties, but large variability of gauge weights among split sample (Figure 6) hints in that direction.

Specific comments

Abstract Line 6-7: According to Line 210 of the Result Section 15.3% and 7.1% correspond to the improvement when only 5 out of 7 catchments are considered.

Line 38-39: It seems for me that the main focus of the study is developing a new method for rain gauge weighing rather than determining a superior interpolation method since only one existing method (i.e., IDW) was examined.

Line 57-60: What about modeling approaches when precipitation correction factor is used as calibration parameter. I think it will be advantageous to mention these techniques here to highlight the novelty of the proposed method.

[Figure]

Line 64-66: Despite what is claimed here I found no analysis of potential influence of catchment characteristics on performance of the proposed method.

Section 2: No information on used temperature data is provided, which I assume is necessary to run hydrological models used in this study

Line 69: What method was applied to derive monthly PET? Is that a better choice to rely on monthly to hourly disaggregated values than to compute a simple temperature-based PET from hourly temperature data directly?

Line 74-75: Here readers are referred to Table 1 for catchment properties. However, Table 1 only provides catchment size. Consider change the wording in these Lines or provide more catchment properties in Table 1.

Line 79-97: Some details regarding catchments (e.g., when the worst flood has occurred or number of recent floods) are not relevant for the performed analysis. On the other hand, additional information regarding catchment physiography, such as, elevation range, mean annual precipitation etc. would be useful for understanding the difference among the catchments and will give a chance to put the findings of the study in the context of different hydrological conditions. Moreover, instead of describing geographical location of the catchments consider providing a map of study catchments. Please indicate the role of possible anthropogenic influence on streamflow simulation in these catchments.

Line 99-101: Please provide a rationale on selecting these three models. Are any of them used in operational flood forecast?

Line 116: The value of power parameter is chosen arbitrary. Please provide a rationale for this value.

Line 121-135: This portion is difficult to understand. Consider revising it. Please provide more details on optimization procedure for weights definition. Was it a part of calibration procedure, where all the weights were identified simultaneously with model

parameters? Where do coefficients (e.g., 2.03 and -4.74) in equation 4 come from? Please provide an explanation. Finally, in this section you refer to likelihood function that explained only later in Section 3.3.1 and is specific to the selected optimization algorithm. Can the proposed gauge weighing method be used in the context of optimization algorithms that rely on maximizing objective function instead of likelihood function?

Section 3.2: Please provide more details regarding set up of hydrological models (i.e., lumped, semi-distributed, distributed).

Line 145-146: Please explain in more detail what is "catchment exchange term". Is it one of the calibrating parameters?

Line 175-179: Why RMSE was chosen as a performance measure instead of common metrics such as NSE or KGE?

Table 2, Table 3: consider merging Table 2 and 3 to ease comparison between optimization and evaluation periods described in Lines 181-192. Consider change the name of the column "Opt" to OGW to be consistent with the text.

Line 185-187: I did not understand this sentence.

Line 189: Where can the reader see that IDW rainfall for GR4H model had the lowest RMSE for 5 out of 7 catchments? According to Table 2 in optimization period 6 out of 7 catchments had lower RMSE for IDW than for OGW. According to Table 3 for evaluation period none of 7 catchments had smaller RMSE for OGW.

Line 190-193 and Line 210: Is it feasible to compare the performance of the best IWD to the best OGW case disregarding of the model used? According to Introduction (Line 48-50) the success of interpolation methods is greatly influenced by the choice of hydrological models.

Line 193-203: Is the proposed method is at all feasible for the conditions when gaps are present in the data? Is it possible that the largest weight will be assigned to the gauge

with data gap that covers the evaluation period? In this case it is likely to deteriorate model performance in the evaluation period that could have been compensated by the information from other gauges in the traditional interpolation methods.

Line 203-206: Is it possible to define/suggest a minimum rain gauge density required for the successful application of the proposed method?

Section 4.1: Why Paddys Flat was selected as a case study?

Figure 3: Is this Figure necessary? The Figure simply shows observed streamflow and IDW interpolated rainfall for the whole study period. It would be more informative if apart from observed discharge it will display discharge simulated by different models with corresponding OGW rainfall for a selected period so that discharge would be actually visible on the Figure.

Line 227: OGW leads to increase of cumulative rainfall for HBV and PDM for all split samples, however for some split samples of GR4H the cumulative rainfall is similar to the IDW rainfall.

Line 233-235: These findings are based on Paddys Flat that according to the Figure 2 did not have large change of rainfall amount compared to IDW. Are these results similar for Tully catchment where the difference between OGW and IDW rainfall amounts was larger than 1000 mm?

Line 236, Line 281: The developed method identifies gauges that add value for the streamflow simulation. This manuscript did not investigate the value of identified gauges for the flood forecast.

Figure 7: Instead of average value of parameters for OGW please provide a box plot of parameter values resulting from each split sample to evaluate the stability of parameters. The decimal point on y axis is missing.

Line 244-250, Line 258-260: Since the model parameters were not introduced for each model in the Methods Section, please specify which parameter mentioned here and

are responsible for soil moisture, evapotranspiration, percolation, fast flow, base flow, catchment exchange parameter etc. to make interpretation of Figure 7 easier.

Line 272-273: This study does not investigate which model structure is likely to benefit more from inclusion of soil moisture for calibration or assimilation purposes. Consider adding citation or revising this sentence.

Line 269-270, 275-278 and Figure 9: Figure 9 is not appropriate to make any conclusions regarding IDW and OGW performance for different streamflow events. On this Figure individual events cannot be identified unambiguously and it is hard to say which IGW event correspond to which OGW event. Consider modifying this Figure by selecting several different streamflow events and showing performance of different gauge weighing methods and models for them.

Line 279: Add the results for all study catchments as supplementary material to prove this point.

Line 287-289: Is it because of inability of GR4H to represent the internal dynamics of the catchments or is it because the catchment exchange parameter accounts and corrects for possible bias present in the input data making adjustment of gauge weight (that in essence also corrects for input data bias) redundant?

Technical corrections

Line 23 and 37: Using "QC'd" instead of "quality controlled" is confusing. In general, consider spelling "quality control" instead using QC. It is only used 4 times in the paper.

Line 41: "Thiessen polygons" instead of "Theissen's polygons"

Line 46: Abbreviations OK and PCRR are not used further in the text and therefore can be omitted.

Line 49: "is" instead of "iss"

Line 82: Abbreviation SA is redundant as it is not used further in the text

Line 87 and 93: Abbreviation NSW is redundant as it is not used further in the text

Line 95: "Queensland" instead of "Queenslad"

Line 106: "operator" instead of "oeprator"

Line 107: "an" instead of "on"

Figure 1: Decimal point on y axis is shifted

Figure 4: Label the different split samples according to the period they were calibrated on

References

Anctil, F., Lauzon, N., Andréassian, V., Oudin, L., Perrin, C., 2006. Improvement of rainfall-runoff forecasts through mean areal rainfall optimization. J. Hydrol. 328, 717–725. https://doi.org/10.1016/j.jhydrol.2006.01.016

Duethmann, D., Zimmer, J., Gafurov, a., Güntner, a., Kriegel, D., Merz, B., Vorogushyn, S., 2013. Evaluation of areal precipitation estimates based on downscaled reanalysis and station data by hydrological modelling. Hydrol. Earth Syst. Sci. 17, 2415–2434. https://doi.org/10.5194/hess-17-2415-2013

Heistermann, M., Kneis, D., 2011. Benchmarking quantitative precipitation estimation by conceptual rainfall-runoff modeling. Water Resour. Res. 47, 1–23. https://doi.org/10.1029/2010WR009153

Schaefli, B., Talamba, D.B., Musy, A., 2007. Quantifying hydrological modeling errors through a mixture of normal distributions. J. Hydrol. 332, 303–315. https://doi.org/10.1016/j.jhydrol.2006.07.005

Yilmaz, K.K., Hogue, T.S., Hsu, K., Sorooshian, S., Gupta, H. V., Wagener, T., 2005. Intercomparison of Rain Gauge, Radar, and Satellite-Based Precipitation Estimates with Emphasis on Hydrologic Forecasting. J. Hydrometeorol. 6, 497–517.

[Figure]

https://doi.org/10.1175/JHM431.1

---

## Author Comment (AC1) · 14 Nov 2019

The remarks made by the Reviewers and the Associate Editor are written in italics, and the replies in normal font and blue color. In the citation of the remarks, the original page, paragraph, table, figure and line numbers have been used. However, in the replies these numbers refer to the numbers in the revised manuscript unless otherwise specified.

*Reviewer #1*

*The study introduces a new methodology to optimise gauge weights as a part of hydrologic model calibration processes. The study also shows that areal rainfall estimation obtained from the new optimisation led to improved performance of rainfall-runoff models, compared to the traditional way (i.e., IDW). This is an interesting approach to produce optimised rainfall forcing for better hydrologic modeling. However, there are some places in the manuscript that need further clarification.*

*Major comments:*

*Extrapolation capacity of model parameters The new optimisation methodology suggested in the study decides gauge weightings based on current gauge configuration and the proportion of available observations (eq. 4) and the optimisation method affects model parameter values (as shown in Figure 7). In this context, can we guarantee that the model parameters perform well also with areal rainfall estimated from other rainfall data sources (e.g., reanalysis for future projection), compared to the model parameters obtained from the IDW? Moreover, when the gauge configuration is changed (e.g., more gauges are installed in the future), how can we use the obtained model parameters & gauge weightings? Do we have to calibrate models again with the new gauge set-up? Can you address this point with more details?*

The model will need to be optimized to account for any alterations to forcing or calibration data. This is now addressed on P6L156

"Similar to the IDW method, any alterations to forcing data via re-analysis or inclusion of additional gauges will accordingly require the optimization process to be revised."

*Specific comments*

*p.3 l.69: why PET is considered here? Is it one of forcing data for models? Please clarify.*

Yes PET is used as forcing data for the models. This is now addressed on P3L90

"Rainfall and potential evapotranspiration (PET) data are used to force the models."

*p.3/Table3: can you specify what time period you considered for each catchment? i.e., the length of 100% time used to calculate % of time that no gauge has an observation.*

We assume table 1 in the original manuscript is referred to and not Table 3 of the original manuscript as stated.

The caption for Table 2 has been amended to include. "Statistics shown are for the collected rainfall record beginning 1st of January 2007 and ending 31st of December 2013, except for in the Onkaparinga and South Esk catchments where rainfall was first recorded on 1st of November 2007"

P3L90  The description for the data set has been amended to include "With the exception being that rainfall in the Onkaparinga and South Esk catchments was first recorded on 1st of November 2007"

Additionally the description for the evaluation strategy has been amended P7L196 to include.

"A warm-up period of one year beginning 1st of January 2007 and ending 31st of December 2008 was used. In the Onkaparinga and South Esk catchments a warm-up period of two months was used."

*p.5 eq.4: how do other constant values are decided? Do they have any (physical) meaning for the calculation?*

This is now addressed on P6L155 as follows

"The constants in equation 4 can be modified to allow different parameter spaces to be searched for each gauge, that has a rainfall time series which is more or less complete than the surrounding gauges."

*p.7 l.175: Why don't you consider more performance metrics (e.g., correlation, bias of annual runoff) to evaluate the performance of IDW vs OGW from various aspects?*

Both bias and correlation are considered in the RMSE. This is now addressed on P7L200 "The Root Mean Square Error (RMSE) was chosen to evaluate model simulations for widespread applicability to explain simulation trends such as bias, standard deviation, and correlation.".

*p.8 l.219-220: this point should be more carefully addressed. This also could mean that the model parameters are forced, to some extent, to describe runoff better, regardless of other hydrologic variables (i.e., over-fitting to runoff).*

This is now addressed on P9L253

"It is common for hydrologists to link the improved simulation performance and increase in the number of parameters with over-fitting. The consistent improvement in simulation skill in the evaluation periods for the PDM and HBV model demonstrates that over-fitting does not occur for these models. However, the consistent decrease in simulation skill in the evaluation periods for the GR4H model demonstrates that over-fitting did occur for this model. By testing different gauge weightings the OGW methodology takes greater advantage of the available rainfall time series when compared to the IDW methodology. As such it is hypothesized that the GR4H model is not able to take advantage of the additional data and is subject to over-fitting."

*And on* P10L293

"This lack of change in parameters for GR4H indicates that the model is not taking advantage of the additional data."

*Figure 1. How did you deal with the time steps with missing observation, especially for the Onkaparinga catchment where most of gauges have less than 50% of time series during the study period, so probably there are some time steps when all gauges are not available?*

This is now addressed on P4L97

"When no gauge within the catchment had an observation, the rainfall is assumed to be zero."

Further, Table 1 indicates how often no data are available. Which is a feature for all catchments.

*Figure 2 vs Figure 4: given that HBV and PDM shows very similar results in Figure 4 (i.e., OGW increased rainfall), I would expect similar results between the HBV and PDM also from Figure 2; do you have any idea why both figures are not consistent?*

Thank you for noticing this. There was an error in the labelling of figure 3. The columns for South Esk and Paddy's Flat were in the wrong location.

*Figure 7: is it showing the average of parameters obtained from the six split samples? Do you also check the "spread" of parameters (e.g., interquartile range) between IDW and OGW?*

Figure 8 now shows the parameters from all split samples. Further, an error was made in the original manuscript and the old figure indicated the parameters for Warwick.

***Minor comments***

*p.3: you may want to consider providing the map of study catchments*

A catchment locality map is now shown in figure 1

*p.2 l.49: iss -> is*

Corrected.

*p.3 l.69: please explain "PET"*

This is now addressed on P3L90

"Rainfall and potential evapotranspiration (PET) data are used to force the models."

*p.4 l.107: on observation -> an observation*

Corrected.

*p.9 l.260: which one is the catchment exchange parameter in Figure 7?*

This is now addressed on P6L169 *by adapting the following line.*

"A potential problem with the GR4H model is that the catchment exchange parameter X2 allows for either the import or export of water into the model."

*Figure 1: please make the y-axis label clear, e.g., 0.2 and 0.3 look like 02 and 03, respectively.*

This is now addressed in figure 2

 *Figure 2: can you make it clear that here the change mean "OGW-IDW"?*

This is now addressed in figure 3

*The y-axis label now reads*

*"Mean change in rainfall between OGW and IDW [mm]"*

*Figure 4: this is for the Paddys Flat, right? Please clarify.*

The captions for Figures 5-10 are updated to outline that they are for Paddy's Flat.

*Figure 8: its't the X-axis from Jan 2008 to Dec 2013?*

Yes that is correct. The x-axis label for figure 9 has been updated.

ashley.wright@monash.edu

[revised manuscript text omitted]

35 streamflow. For their study catchment Xu et al. (2013) found that hydrologic simulation skill improved significantly until a gauge density of 0.4 gauges per 1000 km$^2$ was reached and that simulation skill degraded for rain gauge densities larger than 1.4 gauges per km$^2$. Through random sampling Anctil et al. (2006) explored the impact that different rainfall gauge weightings had on streamflow forecasts and determined that the best performing combination of rain gauges came from a sub-sample of the available rain gauges. These studies either indicate that systematic errors have not been sufficiently quality controlled

40 or that the methodology to interpolate areal rainfall from the gauge network does not adequately weight important gauges. Therefore this study focuses on developing an interpolation method that adequately weights important rainfall gauges.

There are a variety of interpolation methods used to obtain an areal rainfall estimate from a gauged network ranging from simple methods such as the Thiessen's polygons and inverse distance weighting (IDW) methods, to more complex methods such as kriging. In a review of different methods for the spatial interpolation of rainfall data Ly et al. (2013) found no optimal

45 spatial interpolation method for operational hydrology, and that interpolation performance often depended on the density of the gauge network, the spatial and temporal resolution of the data, and parameters such as those used by the semi-variogram in kriging. Mair and Fares (2011); Chen et al. (2017), and Liu et al. (2018) compared interpolation methods based on their ability to simulate streamflow, with each study using a different hydrologic model and catchment. In respective order they found that ordinary kriging, principal component regression with residual correction, and IDW interpolation of gauged rainfall data

50 yielded superior streamflow simulations. However, none of the studies included all three methodologies. The use of a single but different catchment and single but different hydrologic model for each study makes it difficult to determine the robustness of each methodology. It is hypothesized that the best performing interpolation methods are not optimized for hydrologic simulations and that their success is likely to be greatly influenced by the choice of hydrologic model, chosen catchment, and simulation metric.

55 The search for an optimal interpolation of rainfall gauges for hydrologic simulation is essentially an inverse modeling question that asks, how do hydrologists extract optimal rainfall forcings from streamflow simulations? Methods to extract optimal rainfall from streamflow range from those that do not need rainfall forcing such as a cascading linear inversion of runoff (Hino, 1986) to those that require rainfall forcing such as linear inversion (Kirchner, 2009), optimization of storm multipliers (Kavetski et al., 2006; Vrugt et al., 2008), reduction of forcing data (Wright et al., 2017), and the inclusion of

60 ancillary data (Wright et al., 2018). Further Schaefli et al. (2007) and Duethmann et al. (2013) use precipitation correction

factors that are suitable for mountainous or data sparse regions. The nature of these studies implies that rainfall obtained from interpolation methods is not *true* rainfall. However, no example is found in the literature which presents a methodology to optimally interpolate rainfall gauge data for hydrologic simulations and provide insights derived therein. To develop an algorithm for the optimization of gauge weights (OGW) that allows the estimation of rainfall which optimizes hydrologic simulation skill it is necessary to include additional degrees of freedom in the parameter optimization process. Comparisons of streamflow forecast skill when forced with varying quantitative precipitation estimates (Heistermann and Kneis, 2011), and rain gauge, radar, and satellite based precipitation estimates (Yilmaz et al., 2005), demonstrate that calibration may obscure error and bias in rainfall estimates, making it essential that rainfall estimates and their resulting calibrated parameters be thoroughly evaluated outside of the optimization period. Many hydrologic purists are strong advocates of *Occam's Razor* and are, rightfully so, concerned that adding hydrologic parameters will result in over-fitting of hydrologic models. Keeping this in mind the value of additional parameters should be determined through the ability of those parameters to provide improved solutions for hydrologic simulations in either an evaluation or forecast period.

For the development of a robust methodology to optimally interpolate rainfall gauges it is essential to consider a variety of; catchments, rainfall-runoff models, rainfall interpolation schemes, simulation metrics, and quality of data. This study develops a methodology to optimally interpolate rainfall gauges for streamflow simulation using a variety of catchments, rainfall runoff models and quality of data. A review of different methods for the spatial interpolation of rainfall data by Ly et al. (2013) found that no spatial interpolation methodology consistently provided the best streamflow simulations. Consequently streamflow simulations forced by OGW rainfall are benchmarked against those obtained with IDW rainfall and are deemed suitable if there is a consistent improvement noticed. Situations in which the OGW is not appropriate are highlighted. Lastly, the OGW methodology is part of the optimization process. As such using an alternate simulation metric would still yield the OGW for that metric.

This paper contributes to the hydrologic sciences by developing a novel OGW method which is used in the estimation of areal rainfall for hydrologic simulations and providing insights on the potential influences that catchment characteristics and the choice of hydrologic model pose for areal rainfall estimates based on gauge interpolation.

**2 Data set**

With locations shown in Figure 1 this study uses seven Australian catchments that are susceptible to major flooding, and having different rainfall gauge densities, climatology, area, response times, flow characteristics, and seasonality of rainfall. Further, data availability was also an important factor in choosing the catchments. Data are collected for the 7 year period beginning $1^{st}$ of January 2007 and ending $31^{st}$ of December 2013. With the exception being that rainfall in the Onkaparinga and South Esk catchments was first recorded on $1^{st}$ of November 2007. Rainfall and potential evapotranspiration (PET) data are used to force the models. Monthly PET data from the Australian Water Availability Project (AWAP) (Raupach et al., 2009, 2012) were used at and disaggregated to a uniform hourly rate for each month. This disaggregation strategy is appropriate since streamflow simulations are relatively insensitive to the temporal resolution of PET (Samain and Pauwels, 2013). Both hourly rainfall and

streamflow data (Bureau of Meteorology, 2019b) were obtained from the Bureau of Meteorology (BoM) with the catchments
being delineated using the Australian geofabric data set (Bureau of Meteorology, 2019a). All rainfall gauges within 10 km of
the catchment boundary were considered. A summary of the catchment properties is given in Table 1, whilst an indication of
the available observations for each rain gauge in each catchment is given in Figure 2. When no gauge within the catchment had
an observation, rainfall is assumed to be zero. Ephemeral streams were defined as those having zero flows in $> 4\%$ of their
records (Bennett et al., 2016), and Köppen Geiger climate classifications were taken from Peel et al. (2007). Brief summaries
of each catchments main characteristics are as follows:

[revised manuscript text omitted]

where n is the total number of gauges used and $p_j$ is the proportion of observations available for gauge j. The constants in equation 4 can be modified to allow different parameter spaces to be searched for each gauge, that has a rainfall time series which is more or less complete than the surrounding gauges. Further, $w_j^{max}$ were restricted to values that ensured a maximum scaled weighting of 0.55 if that gauge had all observations available. Similar to the IDW method, any alterations to forcing data via re-analysis products or inclusion of additional gauges will accordingly require the optimization process to be revised.

**3.2 Hydrologic models**

The hydrologic models used in this study were chosen based on their proven performance, contrasting routing schemes, and complexity. A description of the parameters and their ranges can be found in Table 2. Both the modèle de Génie Rural à 4 paramètres Horaire (GR4H) and Hydroliska Byråns Vattenbalansavdelning (HBV) model use unit hydrograph (UH) routing whilst the Probability Distributed Model (PDM) uses a dynamic storage based routing approach. Both HBV and PDM close the water balance whilst GR4H does not (Perrin et al., 2003). Each catchment is modelled in a lumped fashion.

**3.2.1 GR4H**

The GR4H is an hourly application of the modèle de Génie Rural à 4 paramètres Journalier (GR4J) model (Perrin et al., 2003) which utilizes two state variables, soil water storage and routing water storage. The model is mathematically parsimonious and consists of 4 parameters which, along with the state variables, describe evapotranspiration, percolation, and both slow and fast runoff. A potential problem with the GR4H model is that the catchment exchange parameter X2 allows for both the import or export of water into the model. Thus the water balance is not closed and potential biases in observation data are not able to be corrected.

**3.2.2 PDM**

The PDM as described by Moore (2007) consists of a set of functions used to describe various hydrologic systems. The model consists of four states, a cascade of two linear stores which are used to describe surface runoff, a linear store used to describe subsurface flow and a catchment based soil moisture store that is considered to consist of soil moisture stores with varying capacities that are able to be represented by a Pareto distribution. The PDM uses 9 model parameters and routing is a dynamic storage based process.

**3.2.3 HBV**

Of the many HBV model (Lindström et al., 1997) variants, this paper uses the version developed by Matgen et al. (2006) and used by Pauwels and De Lannoy (2015). The model consists of three state variables, a soil reservoir, a slow reservoir,

and a fast reservoir. The 11 parameters, 3 state variables, and subsequent governing equations are used to calculate the actual evapotranspiration, infiltration, effective rainfall, percolation, and proportion of effective rainfall that enters the fast and slow reservoirs, outflow from each reservoir, and UH based routing.

**3.3  Parameter estimation**

The estimation of model and OGW parameters was performed simultaneously using the Differential Evolution Adaptive Metropolis algorithm with past state sampling and snooker updating (DREAM$_{ZS}$) of Vrugt (2016). The default settings outlined in Vrugt (2016) were used. DREAM$_{ZS}$ finds the posterior parameter distribution which maximizes a chosen likelihood function.

**3.3.1  Likelihood function**

The optimal weighting of rainfall gauges will produce rainfall specific to the chosen likelihood function and will change if a likelihood function which places priority on low flows is chosen instead of a likelihood function that places priority on high flows. The same can be said if objective functions are maximized instead of likelihood functions. For this study a Gaussian likelihood function (Thiemann et al., 2001) is used.

**3.4  Evaluation strategy**

The evaluation of optimized parameters and gauge weightings was carried out via split sample testing. A warm-up period of one year beginning $1^{st}$ of January 2007 and ending $31^{st}$ of December 2008 was used. In the Onkaparinga and South Esk catchments a warm-up period of two months was used. The skill of streamflow simulations was then evaluated for the six year period beginning $1^{st}$ of January 2008 and ending $31^{st}$ of December 2013. The optimization and evaluation periods were 5 and 1 years long respectively. The optimization and evaluation periods were alternated until all 6 years of evaluation data had been used for evaluation. This resulted in a total of 6 split samples for each catchment.

The Root Mean Square Error (RMSE) was chosen to evaluate model simulations for widespread applicability to explain simulation trends such as bias, standard deviation, and correlation. RMSE is given by

$$\text{RMSE} = \sqrt{\frac{\sum_{i=1}^{t}(q_i^s - q_i^o)^2}{t}}, \tag{5}$$

for which t represents the length of the optimized simulation period, and $q_i^s$, and $q_i^o$, represent the simulated, and observed streamflow at time step i respectively. Relative improvements in simulation skill are determined by calculating the ratio of RMSE between the stated cases. Lastly, to provide a comparison with rainfall observations, streamflow is converted to units of mm/h. This calculation involves dividing the streamflow by the catchment area.

**4 Results and discussion**

210    The results of the split sample testing are presented as an RMSE average of the samples for each catchment and model for the optimization and evaluation periods in Tables 3 and 4 respectively. The HBV model consistently yields lower RMSE values for the optimization period using both the IDW and OGW rainfall products. As expected, the addition of extra parameters for the optimized rainfall product consistently led to notable improvements throughout the optimization period. There are however a few split samples in which the additional parameters used in the OGW methodology led to a marginal increase in RMSE.

215    This can happen when the scaling of gauge weightings, as described in 3.1.2, does not allow the optimization process to attain the same gauge weights as the IDW method. A possible scenario in which this could happen occurs when the IDW method inappropriately applies a weighting larger than $55\%$ to a single gauge.

     Whilst the OGW rainfall estimates generally led to improvements in the evaluation period, there was much less consistency in performance between models. For IDW rainfall the GR4H model had the lowest RMSE for 2 out of the 7 catchments. The

220 best simulations were obtained for 5 out of the 7 catchments when OGW rainfall was used. On average the observed relative improvements in the evaluation period were $-7.9\%$, $6.2\%$, and $8.3\%$ for the GR4H, HBV, and PDM models, respectively. If the best IDW simulation for all models is compared to the best OGW for all models simulation an average improvement of $4.3\%$ was observed.

     To uncover the reasons for the apparent mismatch in performance between the optimization and evaluation periods it is

225 necessary to reflect on the methodology and available data. The two catchments in which the IDW simulations performed better than the OGW simulations both have deficiencies in data availability. Table 1 and Figure 2 show that the Onkaparinga catchment has 21 gauges, only one of which has data available for more than $55\%$ of the time. Since they are more able to contribute to an increase in simulation skill in the optimization period, gauges with a larger proportion of observations available in the optimization period are more likely to obtain higher weights. If these gauges have an overall low proportion

230 of observations available they are more likely to not have observations available in the evaluation period. This is then likely to lead to an increase in simulation skill for the optimization period but not for the evaluation period. This improvement in skill in the optimization period and decrease in skill in the evaluation period was observed in the Onkaparinga catchment, a catchment with the overwhelming majority of gauges having $< 50\%$ data availability. Consequently, this technique is not recommended for catchments that have a overwhelming majority of rainfall gauges with poor ($< 50\%$) data availability. Aside from the

235 Onkaparinga catchment the catchment of Yatton is the only other catchment in which the IDW rainfall simulations perform better than the OGW simulations in evaluation periods. The Yatton catchment has one gauge per 2806 km$^2$ whereas every other catchment has at least one gauge per 200 km$^2$. Consequently, a minimum gauge density of at least one gauge per 200 km$^2$ is recommend. Zeng et al. (2018) demonstrated that identification of model parameters became increasingly difficult as the density of the rain gauge network decreased. This effect is made worse by the OGWs. Consequently, a poor gauge density

240 does not allow for adequate estimation of hydrologic model parameters, gauge weightings, and areal rainfall. If the results of these two catchments are not considered, comparing the best OGW simulation for all models to the best IDW simulation for all models led to improvements of $15.3\%$ and $7.1\%$ in the optimization and evaluation periods, respectively.

Applying all three models and selecting the best performing model resulted in the OGW leading to considerable improvements. It is however concerning to see that the simulation skill in the evaluation periods consistently decreased for the GR4H model. If some models are not well suited to be used with OGW rainfall estimates then it is imperative to know which elements of the model structure contribute to this decrease in skill. The average annual change in rainfall and net forcing, where net forcing are defined as rainfall minus evapotranspiration, between IDW and OGW simulations, are shown in Figure 3 and Table 5 respectively. According to the IDW rainfall interpolation the Tully catchment does not observe enough rainfall to explain the observed streamflow. Consequently, it is likely that the OGW method compensates for this by weighting gauges that observe the highest rainfall for the Tully. OGW rainfall is larger than IDW rainfall for all hydrologic models and catchments, except Yatton. It was also common for both HBV and PDM to obtain larger OGW rainfall estimates than GR4H. This trend is in stark contrast to the pattern in the change of net forcing. The mean change in net forcing across catchments was significantly lower for HBV and PDM than it was for GR4H, indicating that the internal dynamics of the HBV and PDM adapt to a change in rainfall by altering evapotranspiration while GR4H does not.

It is common for hydrologists to link the improved simulation performance and increase in the number of parameters with over-fitting. The consistent improvement in simulation skill in the evaluation periods for the PDM and HBV model demonstrates that over-fitting does not occur for these models. However, the consistent decrease in simulation skill in the evaluation periods for the GR4H model demonstrates that over-fitting did occur for this model. By testing different gauge weightings the OGW methodology takes greater advantage of the available rainfall time series when compared to the IDW methodology. As such it is hypothesized that the GR4H model is not able to take advantage of the additional data and is subject to over-fitting.

**4.1 Case Study - Paddys Flat**

This case study was designed to develop a deeper understanding of the impact that the OGW and subsequent areal rainfall estimates had on the hydrologic model simulations and internal dynamics. Paddys Flat was chosen for the case study due to the improvement in streamflow simulation skill in the evaluation period being similar to the mean improvement of streamflow simulation skill in the evaluation period across all 7 catchments. Further, results observed in the Paddys Flat case study are considered to be representative of the OGW methodology. Results and their associated discussion are presented sequentially to describe the impacts on rainfall, model parameters, the water balance, and streamflow. The observed streamflow and IDW areal rainfall estimate for Paddys Flat can be seen in Figure 4.

The comparison of the cumulative rainfall volumes obtained for the split sample simulations for each model at Paddys Flat is shown in Figure 5 demonstrating that the OGW tends to lead to an increase in cumulative rainfall volumes. Each split sample obtained through model evaluation is represented by a different line. The cumulative rainfall volumes estimated by GR4H are less than those estimated by HBV and PDM. Each split sample produced different behavior according to the data set it was calibrated against. It is expected that results would become more homogenous when longer data sets are used for optimization. However, it is necessary to determine if there is a tendency for low or high magnitude rainfall observations to be estimated differently by the OGW or IDW interpolation methods.

[revised manuscript text omitted]

The OGW estimation of areal rainfall enabled both HBV and PDM to simulate internal dynamics with improved consistency. Models that have more consistency in their representation of internal dynamics are better positioned to benefit from the inclusion of soil moisture data for calibration and/or assimilation purposes (Li et al., 2016). Similar benefits are likely to be observed when updating internal states through the assimilation of observed streamflow. The relative impact of the OGW methodology on streamflow, when compared to the IDW methodology, can be observed in Figure 10. Events can be distinguished by lines of hysteresis and the observed streamflow. For all three models there are a number of flood events that are not adequately simulated. It is hypothesized that this occurs as a result of insufficient rainfall being observed to simulate the observed streamflow. It is clear that for larger magnitude events the OGW methodology brought the streamflow simulations closer to observations for the HBV and PDM throughout the evaluation period. Conversely, the use of OGW rainfall areal rainfall estimates with GR4H led to marginally improved streamflow simulations for one event and significantly worse streamflow simulations for another event.

**5    Conclusions**

This study developed a methodology to optimally weight rainfall gauges such that improved streamflow simulation skill could be obtained for three different conceptual hydrologic models. The OGW methodology developed was tested on seven Australian catchments and a comparison of streamflow simulations obtained using the IDW and OGW methods demonstrated an improvement in streamflow RMSE of $15.3\%$ and $7.1\%$ in the optimization and evaluation periods respectively, for catchments that have multiple rainfall gauges with observations available $> 50\%$ of the time and more than one rainfall gauge every 200 $\mathrm{km}^2$. The methodology did not work equally well for the three hydrologic models chosen. Improvements in evaluation periods were only noticed for the PDM and HBV hydrologic models, and not for the GR4H hydrologic model. The most likely explanation for this is the inability of the GR4H model to represent internal dynamics that are consistent with a conceptual understanding of the rainfall-runoff process. Further research could explore the impact that OGW has on parameter uncertainty

and possible interactions between OGW and the inclusion of catchment exchange or bias correction parameters. Catchment characteristics were not found to influence the applicability of the OGW methodology. This methodology opens new possibilities for model evaluation, understanding of forcing uncertainty, and data assimilation studies in hydrology.

345 *Author contributions.* AW conducted the experimental work, contributed towards the theory and wrote the paper. JW assisted in the writing process. VP and DR contributed towards the theory and assisted in the writing process.

*Acknowledgements.* The authors would like to extend their gratitude to the anonymous reviewers for their comments and recommendations. This work was supported by the Multi-modal Australian Sciences Imaging and Visualisation Environment (MASSIVE) (http://www.massive.org.au).

[revised manuscript text omitted]

**Table 1.** Catchment area, number of rainfall gauges, the % of time that no gauge had an observation for each of the study catchments, average annual IDW rainfall over the simulation period, and minimum and maximum elevation. It should be noted that IDW rainfall is not regarded as the truth. Statistics shown are for the collected rainfall record beginning 1$^{st}$ of January 2007 and ending 31$^{st}$ of December 2013, except for in the Onkaparinga and South Esk catchments where rainfall was first recorded on 1$^{st}$ of November 2007.

| Catchment | Number of gauges | % of time no gauge works | Catchment area km$^2$ | Annual IDW rainfall [mm] | Minimum elevation [m] | Maximum elevation [m] |
|---|---|---|---|---|---|---|
| Hurdle Creek | 12 | 14.6 | 156 | 582.6 | 165.3 | 1062.4 |
| Onkaparinga | 21 | 12.5 | 208 | 485.4 | 179.0 | 616.6 |
| Paddys Flat | 27 | 2.6 | 3112 | 765.1 | 168.6 | 1300.6 |
| South Esk | 13 | 6.902 | 2289 | 616.2 | 177.5 | 1527.3 |
| Tully | 7 | 4.1 | 1390 | 2133.2 | 6.7 | 1198.6 |
| Warwick | 21 | 5.3 | 1379 | 502.1 | 446.2 | 1361.7 |
| Yatton | 7 | 3.6 | 19639.5 | 1015.8 | 99.0 | 1084.9 |

Vrugt, J. A., ter Braak, C. J. F., Clark, M. P., Hyman, J. M., and Robinson, B. A.: Treatment of input uncertainty in hydrologic modeling: Doing hydrology backward with Markov chain Monte Carlo simulation, Water Resour. Res., 44, https://doi.org/10.1029/2007WR006720, 2008.

Wright, A., Walker, J., and Pauwels, V.: Estimating rainfall time series and model parameter distributions using model data reduction and inversion techniques, Water Resour. Res., 53, https://doi.org/10.1002/2017WR020442, 2017.

Wright, A. J., Walker, J. P., and Pauwels, V. R. N.: Identification of Hydrologic Models, Optimized Parameters, and Rainfall Inputs Consistent with In Situ Streamflow and Rainfall and Remotely Sensed Soil Moisture, J. Hydrometeorol., 19, 1305–1320, https://doi.org/10.1175/JHM-D-17-0240.1, 2018.

Xu, H., Xu, C.-Y., Chen, H., Zhang, Z., and Li, L.: Assessing the influence of rain gauge density and distribution on hydrological model performance in a humid region of China, J. Hydrol., 505, 1–12, https://doi.org/10.1016/j.jhydrol.2013.09.004, 2013.

Yilmaz, K. K., Hogue, T. S., Hsu, K.-L., Sorooshian, S., Gupta, H. V., and Wagener, T.: Intercomparison of rain gauge, radar, and satellite-based precipitation estimates with emphasis on hydrologic forecasting, J. Hydrometeorol., 6, 497–517, https://doi.org/10.1175/JHM431.1, 2005.

Zeng, Q., Chen, H., Xu, C.-Y., Jie, M.-X., Chen, J., Guo, S.-L., and Liu, J.: The effect of rain gauge density and distribution on runoff simulation using a lumped hydrological modelling approach, J. Hydrol., 563, 106–122, https://doi.org/10.1016/j.jhydrol.2018.05.058, 2018.

**Table 2.** Parameters and ranges used for the optimization of hydrologic models .

| Model | Parameter | Description | Range | Unit |
|-------|-----------|-------------|-------|------|
| GR4H | X1 | Production store capacity | 1 - 3000 | mm |
| | X2 | Catchment exchange | -27 - 27 | mm |
| | X3 | Routing store capacity | 1 - 660 | mm |
| | X4 | length of unit hydrograph | 1 - 240 | hr |
| HBV | $\lambda$ | Parameter for evapotranspiration | $10^{-2}$ - 3 | - |
| | $S_{max}$ | Capacity of soil reservoir | $1 - 10^3$ | mm |
| | b | Parameter for infiltration | $10^{-2}$ - 3 | - |
| | Pe | Percolation parameter | $10^-2 - 50$ | mm/h |
| | $\alpha$ | Fast reservoir parameter | $10^{-2}$ - 3 | - |
| | $\beta$ | Percolation parameter | $10^{-2}$ - 3 | - |
| | $S2_{max}$ | Capacity of fast reservoir | 1 - 2000 | mm |
| | $\kappa_2$ | Parameter for outflow from fast reservoir | $10^{-1} - 10^2$ | $mm^3/h$ |
| | $\psi$ | Parameter for outflow from fast reservoir | $10^{-1}$ - 3 | - |
| | $\kappa_1$ | Parameter for outflow from slow reservoir | $10^{-3}$ - 1 | $mm^2/h$ |
| | UH | length of unit hydrograph | 1 - 240 | h |
| PDM | $C_{max}$ | Maximum store capacity | 1 - 500 | mm |
| | b | Parameter that controls spatial variability of $C_{max}$ | $10^{-3}$ - 1.8 | - |
| | $b_e$ | Actual evaporation exponent | $0.1 - 5$ | - |
| | $b_g$ | Recharge function exponent | $0.2 - 6.7$ | - |
| | $k_b$ | Baseflow constant | 1 - 2000 | $h/mm^2$ |
| | $C_{min}$ | Minimum store capacity | $(0 - 1) \times C_{max}$ | mm |
| | $S_t$ | Soil tension storage capacity | $(0 - 1) \times C_{max}$ | mm |
| | $\kappa_1$ | Time constant for linear reservoir | 1 - 300 | h |
| | $\kappa_2/$ | Time constant for linear reservoir | $(10^{-6} - 100) \times k_1$ | h |

[Figure]

**Figure 1.** The location of catchments used in this study.

**Table 3.** Average RMSE [mm/h] for simulated streamflow in the optimization periods for each catchment using IDW and OGW rainfall. Values in bold indicate the lowest obtained RMSE.

| Catchment | GR4H | | HBV | | PDM | |
|---|---|---|---|---|---|---|
| | IDW | OGW | IDW | OGW | IDW | OGW |
| Hurdle Creek | 0.033 | 0.031 | 0.032 | **0.028** | 0.033 | 0.028 |
| Onkaparinga | 0.040 | 0.039 | 0.037 | **0.037** | 0.039 | 0.039 |
| Paddys Flat | 0.068 | 0.053 | 0.061 | **0.048** | 0.084 | 0.069 |
| South Esk | 0.037 | 0.033 | 0.035 | **0.032** | 0.043 | 0.039 |
| Tully | 0.175 | 0.177 | 0.165 | **0.151** | 0.170 | 0.157 |
| Warwick | 0.032 | 0.023 | 0.032 | **0.023** | 0.047 | 0.038 |
| Yatton | 0.062 | 0.056 | 0.053 | **0.048** | 0.061 | 0.057 |

[Figure]

**Figure 2.** Box plots showing the median, $25^{th}$ and $75^{th}$ percentile, end points and outliers that describe the proportion of observations available for gauges within each catchment.

**Table 4.** Average RMSE [mm/h] for simulated streamflow in evaluation periods for each catchment using IDW and OGW rainfall. Values in bold indicate the lowest obtained RMSE.

|  | GR4H | | HBV | | PDM | |
| --- | --- | --- | --- | --- | --- | --- |
| Catchment | IDW | OGW | IDW | OGW | IDW | OGW |
| Hurdle Creek | 0.033 | 0.033 | 0.041 | 0.036 | 0.037 | **0.029** |
| Onkaparinga | **0.035** | 0.036 | 0.038 | 0.039 | 0.038 | 0.039 |
| Paddys Flat | 0.083 | 0.103 | 0.082 | **0.077** | 0.093 | 0.080 |
| South Esk | 0.037 | **0.037** | 0.044 | 0.043 | 0.053 | 0.044 |
| Tully | 0.176 | 0.184 | 0.161 | **0.151** | 0.167 | 0.156 |
| Warwick | 0.035 | 0.038 | 0.042 | **0.031** | 0.050 | 0.041 |
| Yatton | **0.068** | 0.075 | 0.069 | 0.072 | 0.068 | 0.078 |

[Figure]

**Figure 3.** The average annual change in estimated areal rainfall for the simulation period between the OGW and IDW areal rainfall estimation methods.

**Table 5.** Average annual change [mm] in rainfall minus evapotranspiration between the IDW and OGW rainfall simulations. A positive change indicates that net model forcings were higher for the OGW simulations.

|  | GR4H | HBV | PDM |
|---|---|---|---|
| Hurdle Creek | 40.0 | 22.9 | 19.8 |
| Onkaparinga | 5.1 | 0.57 | 6.3 |
| Paddys Flat | -13.0 | 3.5 | -33.3 |
| South Esk | 54.5 | -5.3 | 16.6 |
| Tully | 789.3 | 201.8 | 189.2 |
| Warwick | 9.4 | 2.6 | 14.0 |
| Yatton | 115.5 | -11.6 | 22.5 |
| Mean | 143.0 | 60.1 | 33.6 |

[Figure]

**Figure 4.** Observed streamflow and IDW areal rainfall estimates for Paddys Flat.

**Figure 5.** Cumulative areal rainfall estimates obtained for each OGW simulation when compared to the cumulative IDW rainfall areal rainfall estimate for Paddy's Flat. 1-1 reference lines are shown in black. Each colour represents a different split sample.

[Figure]

**Figure 6.** A 1-1 comparison of the average OGW areal rainfall estimates with the IDW areal rainfall estimates for each model for Paddy's Flat.

**Figure 7.** A comparison of the IDW gauge weights, as indicated by a ×, with OGW gauge weights for each split sample simulation for Paddy's Flat. A + indicates a potential outlier.

[Figure]

**Figure 8.** Relative optimized parameters for each rainfall-runoff model using OGW and IDW rainfall estimates for Paddy's Flat. Values of 0 and 1 indicate the lower and upper bounds for each parameter respectively.

[Figure]

**Figure 9.** A representation of each models ability to resolve storage for each split sample and areal rainfall estimation method respectively for Paddy's Flat. Each time series starts at the end of the warm-up period and is adjusted to have the same storage as other samples from the model and rainfall estimation method pair. Each colour represents a different split sample.

[Figure]

**Figure 10.** A comparison of the streamflow simulations for the evaluation periods for each model and areal rainfall estimation method for Paddy's Flat. Events can be distinguished by lines of hysteresis and the observed streamflow. 1-1 reference lines are shown in black.

---

## Author Comment (AC2) · 14 Nov 2019

The remarks made by the Reviewers and the Associate Editor are written in italics, and the replies in normal font and blue color. In the citation of the remarks, the original page, paragraph, table, figure and line numbers have been used. However, in the replies these numbers refer to the numbers in the revised manuscript unless otherwise specified.

*Reviewer #2*

*The authors introduce the OGW (optimization of gauge weights) method and compare their results with the IDW (inverse distance weighting) method, based on data from seven catchments in Australia. Furthermore, results of a case study are presented. This is an interesting study, which deserves to be published – but the presentation quality should be improved. I have just noticed that quite some of my potential comments have already been stated by Reviewer # 1, so I repeat just those, which I find most urgent.*

*General comments:*

(1) *I found the usage units for "streamflow" quite confusing: In Fig.3 it is m3 /s, in Fig. 9 it is mm/h. Is this obtained by dividing the flow in [m3 /s] through the catchment area? Which units should be used in Equation 5 (for q and t)? There is a square root of [streamflow squared, divided by time]. The result can be hardly [mm], as given in Table 2.*

This is now addressed in Figure 4 and table 3. Further on P7L205 the following phrase has been added

"Lastly, to provide a comparison with rainfall observations, streamflow is converted to units of mm/h. This calculation involves dividing the streamflow by the catchment area"

(2) *Please check the references throughout: The "scopus" information is hardly readable, like: "https://www.scopus.com/inward/record.uri?eid=2-s2.0-84884921481{&}doi=10.1016{%}2Fj.jhydrol.2013.09.004{&}partnerID= 40{&}md5=3554bc7f56537b1b95a3a51160ebe40a, 2013." I would regards the DOI information as sufficient.*
The references have been updated to show only the URL where applicable and only the DOI for most references.

*Specific comments:*

(1) *Abstract: "For a selection of 7 Australian catchments this methodology was able to yield improvements of 15.3% and 7.1% .." This is a bit misleading, since you have excluded 2 catchments to get these numbers – right?*

This was not intended to be misleading. It is now addressed on P1L6
"For a selection of 5 Australian catchments"

(3) *Page 2, Line 35: ".. gauge density of 0.4 gauges per km2 ". Please check the numbers. According to Xu et al. (2013) this should be "per 1000 km2 "*
This is now addressed on P2L36
"until a gauge density of 0.4 gauges per 1000 km$^2$"

(4) *P 3, L 68: How did you select the catchments? I understand that all are susceptible to flooding, but are these the catchments, which are most affected?*

The phrase on P3L86 now includes mention of "major flooding". The phrase now reads;

"This study uses seven Australian catchments that are susceptible to major flooding, and having different rainfall gauge densities, climatology, area, response times, flow characteristics, and seasonality of rainfall. Further, data availability was also an important factor in choosing the catchments."

*(5) A map with the locations of the catchments would be very helpful.*

A catchment locality map is now shown in Figure 1

*(6) You should provide some information on average precipitation in the different catchments. This would help to interpret the results in Fig. 2. Values for Tully look intimidating, but I have just found out that Tully (town) is one of the wettest places in Australia.*

Table 1 and its caption has been updated to include the requested information on average annual rainfall.

*(7) P 4, L 116: "taken to be 2 for this study" – why?*

This is now addressed on P5L137

"is an arbitrary power parameter taken to be 2 for this study. This is a commonly applied value , used operationally by the BoM, and consistent with Liu et al (2018)"

*(8) P 6, L 158: Please explain "UH" 2*

The acronym UH is described to be unit hydrograph on P6L162

*(9) Figure 1: Values for Hurdle and Warwick seem to be concentrated around ~ 75 %. Do you have an explanation for that? In general I would have expected at least some values closer to 100 %.*

Values of 75% are considered to be quite high. Due to quality control, maintenance, and equipment malfunction it is unlikely that 100% is ever reached.

*(10) Table 4: You give 3 digits after the decimal point, are those digits significant? With such an uneven distribution, the standard deviation (which is much larger than the mean) is not very meaningful.*

Table 5 now uses 1 digit after the decial and the standard deviation has been removed.

*(11) Table 4: The GR4H values for Tully is 5525 mm – can this be true?*

It is oddly large. But yes it is true. This is now addressed on P9L246

"According to the IDW rainfall interpolation the Tully catchment does not observe enough rainfall to explain the observed streamflow. Consequently, it is likely that the OGW method compensates for this by weighting gauges that observe the highest rainfall for the Tully."

*(12) Equation 6: Shouldn't there be a parenthesis? So you would just be summing up the r values – and the symbol r needs to be explained.*

Equation 6 has been updated. An explanation for *r* is given on P10L299

*(13) Figure 8: One split sample for GR4H IDW shows a totally different behavior. Do you know why?*

A specific reason is not known. A generic reason is given on P10L302

"Each split sample produced different behavior according to the data set it was calibrated against. It is expected that results would become more homogenous when longer data sets are used for optimization."

(13) Figure 4: One split sample for GR4H IDW shows also a different behavior. Do you know why? And is it maybe the same as in Fig. 8?

A specific reason is not known. A generic reason is given on P10L302

"Again each split sample produced different behavior according to the data set it was calibrated against and it is expected that results would become more homogenous when longer data sets are used for optimization."

(14) Figure 9: The PDM model seems to saturate at ~ 2.5 mm/h. Do you have an explanation for this behavior?

This is now addressed on P11L325

"It is hypothesized that this occurs as result of insufficient rainfall being observed to simulate the observed streamflow."

Further tests on the PDM indicate it can simulate larger events but does not due to poor performance in other times of the streamflow record.

**(Some) technical corrections:**

I did not specifically search for typos, but found quite some – so there are probably more. Some examples:

P 1 L 1: hydrolgical Same line: applictions

Corrected

P 2 L 55: The nature …. imply

Corrected

P 3 L 77: Köppen-Gieger (should be "-Geiger").

Corrected

P 4, L 95: Queenslad

Corrected

P 4, L 106: operator

Corrected

P 4, L 127: "to potentially have" should be "do …" right?

No, they potentially have larger weights. It is not definite.

P 5, L 146: "Thus the water balance is not closed and potential biases in observation data corrected for." seems to be incomplete.

Corrected

P 6, L 171: "The skill … were .."

Corrected

*P 10, L 281: "weigh" should be "weight" – right?*

Corrected

ashley.wright@monash.edu

[revised manuscript text omitted]

There are a variety of interpolation methods used to obtain an areal rainfall estimate from a gauged network ranging from simple methods such as the Thiessen's polygons and inverse distance weighting (IDW) methods, to more complex methods such as kriging. In a review of different methods for the spatial interpolation of rainfall data Ly et al. (2013) found no optimal
45  spatial interpolation method for operational hydrology, and that interpolation performance often depended on the density of the gauge network, the spatial and temporal resolution of the data, and parameters such as those used by the semi-variogram in kriging. Mair and Fares (2011); Chen et al. (2017), and Liu et al. (2018) compared interpolation methods based on their ability to simulate streamflow, with each study using a different hydrologic model and catchment. In respective order they found that ordinary kriging, principal component regression with residual correction, and IDW interpolation of gauged rainfall data
50  yielded superior streamflow simulations. However, none of the studies included all three methodologies. The use of a single but different catchment and single but different hydrologic model for each study makes it difficult to determine the robustness of each methodology. It is hypothesized that the best performing interpolation methods are not optimized for hydrologic simulations and that their success is likely to be greatly influenced by the choice of hydrologic model, chosen catchment, and simulation metric.

55  The search for an optimal interpolation of rainfall gauges for hydrologic simulation is essentially an inverse modeling question that asks, how do hydrologists extract optimal rainfall forcings from streamflow simulations? Methods to extract optimal rainfall from streamflow range from those that do not need rainfall forcing such as a cascading linear inversion of runoff (Hino, 1986) to those that require rainfall forcing such as linear inversion (Kirchner, 2009), optimization of storm multipliers (Kavetski et al., 2006; Vrugt et al., 2008), reduction of forcing data (Wright et al., 2017), and the inclusion of
60  ancillary data (Wright et al., 2018). Further Schaefli et al. (2007) and Duethmann et al. (2013) use precipitation correction

factors that are suitable for mountainous or data sparse regions. The nature of these studies implies that rainfall obtained from interpolation methods is not *true* rainfall. However, no example is found in the literature which presents a methodology to optimally interpolate rainfall gauge data for hydrologic simulations and provide insights derived therein. To develop an algorithm for the optimization of gauge weights (OGW) that allows the estimation of rainfall which optimizes hydrologic simulation skill it is necessary to include additional degrees of freedom in the parameter optimization process. Comparisons of streamflow forecast skill when forced with varying quantitative precipitation estimates (Heistermann and Kneis, 2011), and rain gauge, radar, and satellite based precipitation estimates (Yilmaz et al., 2005), demonstrate that calibration may obscure error and bias in rainfall estimates, making it essential that rainfall estimates and their resulting calibrated parameters be thoroughly evaluated outside of the optimization period. Many hydrologic purists are strong advocates of *Occam's Razor* and are, rightfully so, concerned that adding hydrologic parameters will result in over-fitting of hydrologic models. Keeping this in mind the value of additional parameters should be determined through the ability of those parameters to provide improved solutions for hydrologic simulations in either an evaluation or forecast period.

For the development of a robust methodology to optimally interpolate rainfall gauges it is essential to consider a variety of; catchments, rainfall-runoff models, rainfall interpolation schemes, simulation metrics, and quality of data. This study develops a methodology to optimally interpolate rainfall gauges for streamflow simulation using a variety of catchments, rainfall runoff models and quality of data. A review of different methods for the spatial interpolation of rainfall data by Ly et al. (2013) found that no spatial interpolation methodology consistently provided the best streamflow simulations. Consequently streamflow simulations forced by OGW rainfall are benchmarked against those obtained with IDW rainfall and are deemed suitable if there is a consistent improvement noticed. Situations in which the OGW is not appropriate are highlighted. Lastly, the OGW methodology is part of the optimization process. As such using an alternate simulation metric would still yield the OGW for that metric.

This paper contributes to the hydrologic sciences by developing a novel OGW method which is used in the estimation of areal rainfall for hydrologic simulations and providing insights on the potential influences that catchment characteristics and the choice of hydrologic model pose for areal rainfall estimates based on gauge interpolation.

**2  Data set**

With locations shown in Figure 1 this study uses seven Australian catchments that are susceptible to major flooding, and having different rainfall gauge densities, climatology, area, response times, flow characteristics, and seasonality of rainfall. Further, data availability was also an important factor in choosing the catchments. Data are collected for the 7 year period beginning $1^{st}$ of January 2007 and ending $31^{st}$ of December 2013. With the exception being that rainfall in the Onkaparinga and South Esk catchments was first recorded on $1^{st}$ of November 2007. Rainfall and potential evapotranspiration (PET) data are used to force the models. Monthly PET data from the Australian Water Availability Project (AWAP) (Raupach et al., 2009, 2012) were used at and disaggregated to a uniform hourly rate for each month. This disaggregation strategy is appropriate since streamflow simulations are relatively insensitive to the temporal resolution of PET (Samain and Pauwels, 2013). Both hourly rainfall and

streamflow data (Bureau of Meteorology, 2019b) were obtained from the Bureau of Meteorology (BoM) with the catchments
being delineated using the Australian geofabric data set (Bureau of Meteorology, 2019a). All rainfall gauges within 10 km of
the catchment boundary were considered. A summary of the catchment properties is given in Table 1, whilst an indication of
the available observations for each rain gauge in each catchment is given in Figure 2. When no gauge within the catchment had
an observation, rainfall is assumed to be zero. Ephemeral streams were defined as those having zero flows in $> 4\%$ of their
records (Bennett et al., 2016), and Köppen Geiger climate classifications were taken from Peel et al. (2007). Brief summaries
of each catchments main characteristics are as follows:

[revised manuscript text omitted]

155   where n is the total number of gauges used and $p_j$ is the proportion of observations available for gauge j. The constants in equation 4 can be modified to allow different parameter spaces to be searched for each gauge, that has a rainfall time series which is more or less complete than the surrounding gauges. Further, $w_j^{max}$ were restricted to values that ensured a maximum scaled weighting of 0.55 if that gauge had all observations available. Similar to the IDW method, any alterations to forcing data via re-analysis products or inclusion of additional gauges will accordingly require the optimization process to be revised.

**3.2   Hydrologic models**

The hydrologic models used in this study were chosen based on their proven performance, contrasting routing schemes, and complexity. A description of the parameters and their ranges can be found in Table 2. Both the modèle de Génie Rural à 4 paramètres Horaire (GR4H) and Hydroliska Byråns Vattenbalansavdelning (HBV) model use unit hydrograph (UH) routing whilst the Probability Distributed Model (PDM) uses a dynamic storage based routing approach. Both HBV and PDM close
165   the water balance whilst GR4H does not (Perrin et al., 2003). Each catchment is modelled in a lumped fashion.

**3.2.1   GR4H**

The GR4H is an hourly application of the modèle de Génie Rural à 4 paramètres Journalier (GR4J) model (Perrin et al., 2003) which utilizes two state variables, soil water storage and routing water storage. The model is mathematically parsimonious and consists of 4 parameters which, along with the state variables, describe evapotranspiration, percolation, and both slow and fast
170   runoff. A potential problem with the GR4H model is that the catchment exchange parameter X2 allows for both the import or export of water into the model. Thus the water balance is not closed and potential biases in observation data are not able to be corrected.

**3.2.2   PDM**

The PDM as described by Moore (2007) consists of a set of functions used to describe various hydrologic systems. The model
175   consists of four states, a cascade of two linear stores which are used to describe surface runoff, a linear store used to describe subsurface flow and a catchment based soil moisture store that is considered to consist of soil moisture stores with varying capacities that are able to be represented by a Pareto distribution. The PDM uses 9 model parameters and routing is a dynamic storage based process.

**3.2.3   HBV**

180   Of the many HBV model (Lindström et al., 1997) variants, this paper uses the version developed by Matgen et al. (2006) and used by Pauwels and De Lannoy (2015). The model consists of three state variables, a soil reservoir, a slow reservoir,

and a fast reservoir. The 11 parameters, 3 state variables, and subsequent governing equations are used to calculate the actual evapotranspiration, infiltration, effective rainfall, percolation, and proportion of effective rainfall that enters the fast and slow reservoirs, outflow from each reservoir, and UH based routing.

**3.3 Parameter estimation**

The estimation of model and OGW parameters was performed simultaneously using the Differential Evolution Adaptive Metropolis algorithm with past state sampling and snooker updating (DREAM$_{ZS}$) of Vrugt (2016). The default settings outlined in Vrugt (2016) were used. DREAM$_{ZS}$ finds the posterior parameter distribution which maximizes a chosen likelihood function.

**3.3.1 Likelihood function**

The optimal weighting of rainfall gauges will produce rainfall specific to the chosen likelihood function and will change if a likelihood function which places priority on low flows is chosen instead of a likelihood function that places priority on high flows. The same can be said if objective functions are maximized instead of likelihood functions. For this study a Gaussian likelihood function (Thiemann et al., 2001) is used.

**3.4 Evaluation strategy**

The evaluation of optimized parameters and gauge weightings was carried out via split sample testing. A warm-up period of one year beginning $1^{st}$ of January 2007 and ending $31^{st}$ of December 2008 was used. In the Onkaparinga and South Esk catchments a warm-up period of two months was used. The skill of streamflow simulations was then evaluated for the six year period beginning $1^{st}$ of January 2008 and ending $31^{st}$ of December 2013. The optimization and evaluation periods were 5 and 1 years long respectively. The optimization and evaluation periods were alternated until all 6 years of evaluation data had been used for evaluation. This resulted in a total of 6 split samples for each catchment.

The Root Mean Square Error (RMSE) was chosen to evaluate model simulations for widespread applicability to explain simulation trends such as bias, standard deviation, and correlation. RMSE is given by

$$\text{RMSE} = \sqrt{\frac{\sum_{i=1}^{t}(q_i^s - q_i^o)^2}{t}}, \tag{5}$$

[revised manuscript text omitted]

It is common for hydrologists to link the improved simulation performance and increase in the number of parameters with over-fitting. The consistent improvement in simulation skill in the evaluation periods for the PDM and HBV model demonstrates that over-fitting does not occur for these models. However, the consistent decrease in simulation skill in the evaluation periods for the GR4H model demonstrates that over-fitting did occur for this model. By testing different gauge weightings the OGW methodology takes greater advantage of the available rainfall time series when compared to the IDW methodology. As such it is hypothesized that the GR4H model is not able to take advantage of the additional data and is subject to over-fitting.

**4.1 Case Study - Paddys Flat**

This case study was designed to develop a deeper understanding of the impact that the OGW and subsequent areal rainfall estimates had on the hydrologic model simulations and internal dynamics. Paddys Flat was chosen for the case study due to the improvement in streamflow simulation skill in the evaluation period being similar to the mean improvement of streamflow simulation skill in the evaluation period across all 7 catchments. Further, results observed in the Paddys Flat case study are considered to be representative of the OGW methodology. Results and their associated discussion are presented sequentially to describe the impacts on rainfall, model parameters, the water balance, and streamflow. The observed streamflow and IDW areal rainfall estimate for Paddys Flat can be seen in Figure 4.

The comparison of the cumulative rainfall volumes obtained for the split sample simulations for each model at Paddys Flat is shown in Figure 5 demonstrating that the OGW tends to lead to an increase in cumulative rainfall volumes. Each split sample obtained through model evaluation is represented by a different line. The cumulative rainfall volumes estimated by GR4H are less than those estimated by HBV and PDM. Each split sample produced different behavior according to the data set it was calibrated against. It is expected that results would become more homogenous when longer data sets are used for optimization. However, it is necessary to determine if there is a tendency for low or high magnitude rainfall observations to be estimated differently by the OGW or IDW interpolation methods.

[revised manuscript text omitted]

The OGW estimation of areal rainfall enabled both HBV and PDM to simulate internal dynamics with improved consistency.
Models that have more consistency in their representation of internal dynamics are better positioned to benefit from the inclu-
sion of soil moisture data for calibration and/or assimilation purposes (Li et al., 2016). Similar benefits are likely to be observed
when updating internal states through the assimilation of observed streamflow. The relative impact of the OGW methodology
on streamflow, when compared to the IDW methodology, can be observed in Figure 10. Events can be distinguished by lines of
hysteresis and the observed streamflow. For all three models there are a number of flood events that are not adequately simu-
lated. It is hypothesized that this occurs as a result of insufficient rainfall being observed to simulate the observed streamflow. It
is clear that for larger magnitude events the OGW methodology brought the streamflow simulations closer to observations for
the HBV and PDM throughout the evaluation period. Conversely, the use of OGW rainfall areal rainfall estimates with GR4H
led to marginally improved streamflow simulations for one event and significantly worse streamflow simulations for another
event.

**5 Conclusions**

This study developed a methodology to optimally weight rainfall gauges such that improved streamflow simulation skill could
be obtained for three different conceptual hydrologic models. The OGW methodology developed was tested on seven Aus-
tralian catchments and a comparison of streamflow simulations obtained using the IDW and OGW methods demonstrated an
improvement in streamflow RMSE of $15.3\%$ and $7.1\%$ in the optimization and evaluation periods respectively, for catchments
that have multiple rainfall gauges with observations available $> 50\%$ of the time and more than one rainfall gauge every 200
$\text{km}^2$. The methodology did not work equally well for the three hydrologic models chosen. Improvements in evaluation peri-
ods were only noticed for the PDM and HBV hydrologic models, and not for the GR4H hydrologic model. The most likely
explanation for this is the inability of the GR4H model to represent internal dynamics that are consistent with a conceptual
understanding of the rainfall-runoff process. Further research could explore the impact that OGW has on parameter uncertainty

and possible interactions between OGW and the inclusion of catchment exchange or bias correction parameters. Catchment characteristics were not found to influence the applicability of the OGW methodology. This methodology opens new possibilities for model evaluation, understanding of forcing uncertainty, and data assimilation studies in hydrology.

345 *Author contributions.* AW conducted the experimental work, contributed towards the theory and wrote the paper. JW assisted in the writing process. VP and DR contributed towards the theory and assisted in the writing process.

*Acknowledgements.* The authors would like to extend their gratitude to the anonymous reviewers for their comments and recommendations. This work was supported by the Multi-modal Australian Sciences Imaging and Visualisation Environment (MASSIVE) (http://www.massive.org.au).

[revised manuscript text omitted]

**Table 1.** Catchment area, number of rainfall gauges, the % of time that no gauge had an observation for each of the study catchments, average annual IDW rainfall over the simulation period, and minimum and maximum elevation. It should be noted that IDW rainfall is not regarded as the truth. Statistics shown are for the collected rainfall record beginning 1$^{st}$ of January 2007 and ending 31$^{st}$ of December 2013, except for in the Onkaparinga and South Esk catchments where rainfall was first recorded on 1$^{st}$ of November 2007.

| Catchment | Number of gauges | % of time no gauge works | Catchment area km$^2$ | Annual IDW rainfall [mm] | Minimum elevation [m] | Maximum elevation [m] |
|---|---|---|---|---|---|---|
| Hurdle Creek | 12 | 14.6 | 156 | 582.6 | 165.3 | 1062.4 |
| Onkaparinga | 21 | 12.5 | 208 | 485.4 | 179.0 | 616.6 |
| Paddys Flat | 27 | 2.6 | 3112 | 765.1 | 168.6 | 1300.6 |
| South Esk | 13 | 6.902 | 2289 | 616.2 | 177.5 | 1527.3 |
| Tully | 7 | 4.1 | 1390 | 2133.2 | 6.7 | 1198.6 |
| Warwick | 21 | 5.3 | 1379 | 502.1 | 446.2 | 1361.7 |
| Yatton | 7 | 3.6 | 19639.5 | 1015.8 | 99.0 | 1084.9 |

Vrugt, J. A., ter Braak, C. J. F., Clark, M. P., Hyman, J. M., and Robinson, B. A.: Treatment of input uncertainty in hydrologic modeling: Doing hydrology backward with Markov chain Monte Carlo simulation, Water Resour. Res., 44, https://doi.org/10.1029/2007WR006720, 2008.

Wright, A., Walker, J., and Pauwels, V.: Estimating rainfall time series and model parameter distributions using model data reduction and inversion techniques, Water Resour. Res., 53, https://doi.org/10.1002/2017WR020442, 2017.

Wright, A. J., Walker, J. P., and Pauwels, V. R. N.: Identification of Hydrologic Models, Optimized Parameters, and Rainfall Inputs Consistent with In Situ Streamflow and Rainfall and Remotely Sensed Soil Moisture, J. Hydrometeorol., 19, 1305–1320, https://doi.org/10.1175/JHM-D-17-0240.1, 2018.

Xu, H., Xu, C.-Y., Chen, H., Zhang, Z., and Li, L.: Assessing the influence of rain gauge density and distribution on hydrological model performance in a humid region of China, J. Hydrol., 505, 1–12, https://doi.org/10.1016/j.jhydrol.2013.09.004, 2013.

Yilmaz, K. K., Hogue, T. S., Hsu, K.-L., Sorooshian, S., Gupta, H. V., and Wagener, T.: Intercomparison of rain gauge, radar, and satellite-based precipitation estimates with emphasis on hydrologic forecasting, J. Hydrometeorol., 6, 497–517, https://doi.org/10.1175/JHM431.1, 2005.

Zeng, Q., Chen, H., Xu, C.-Y., Jie, M.-X., Chen, J., Guo, S.-L., and Liu, J.: The effect of rain gauge density and distribution on runoff simulation using a lumped hydrological modelling approach, J. Hydrol., 563, 106–122, https://doi.org/10.1016/j.jhydrol.2018.05.058, 2018.

**Table 2.** Parameters and ranges used for the optimization of hydrologic models .

| Model | Parameter | Description | Range | Unit |
|---|---|---|---|---|
| GR4H | X1 | Production store capacity | 1 - 3000 | mm |
| | X2 | Catchment exchange | -27 - 27 | mm |
| | X3 | Routing store capacity | 1 - 660 | mm |
| | X4 | length of unit hydrograph | 1 - 240 | hr |
| HBV | $\lambda$ | Parameter for evapotranspiration | $10^{-2}$ - 3 | - |
| | $S_{max}$ | Capacity of soil reservoir | $1 - 10^3$ | mm |
| | b | Parameter for infiltration | $10^{-2}$ - 3 | - |
| | Pe | Percolation parameter | $10^-2 - 50$ | mm/h |
| | $\alpha$ | Fast reservoir parameter | $10^{-2}$ - 3 | - |
| | $\beta$ | Percolation parameter | $10^{-2}$ - 3 | - |
| | $S2_{max}$ | Capacity of fast reservoir | 1 - 2000 | mm |
| | $\kappa_2$ | Parameter for outflow from fast reservoir | $10^{-1} - 10^2$ | $mm^3/h$ |
| | $\psi$ | Parameter for outflow from fast reservoir | $10^{-1}$ - 3 | - |
| | $\kappa_1$ | Parameter for outflow from slow reservoir | $10^{-3}$ - 1 | $mm^2/h$ |
| | UH | length of unit hydrograph | 1 - 240 | h |
| PDM | $C_{max}$ | Maximum store capacity | 1 - 500 | mm |
| | b | Parameter that controls spatial variability of $C_{max}$ | $10^{-3}$ - 1.8 | - |
| | $b_e$ | Actual evaporation exponent | $0.1 - 5$ | - |
| | $b_g$ | Recharge function exponent | $0.2 - 6.7$ | - |
| | $k_b$ | Baseflow constant | 1 - 2000 | $h/mm^2$ |
| | $C_{min}$ | Minimum store capacity | $(0 - 1) \times C_{max}$ | mm |
| | $S_t$ | Soil tension storage capacity | $(0 - 1) \times C_{max}$ | mm |
| | $\kappa_1$ | Time constant for linear reservoir | 1 - 300 | h |
| | $\kappa_2/$ | Time constant for linear reservoir | $(10^{-6} - 100) \times k_1$ | h |

[Figure]

**Figure 1.** The location of catchments used in this study.

**Table 3.** Average RMSE [mm/h] for simulated streamflow in the optimization periods for each catchment using IDW and OGW rainfall. Values in bold indicate the lowest obtained RMSE.

| Catchment | GR4H | | HBV | | PDM | |
|---|---|---|---|---|---|---|
| | IDW | OGW | IDW | OGW | IDW | OGW |
| Hurdle Creek | 0.033 | 0.031 | 0.032 | **0.028** | 0.033 | 0.028 |
| Onkaparinga | 0.040 | 0.039 | 0.037 | **0.037** | 0.039 | 0.039 |
| Paddys Flat | 0.068 | 0.053 | 0.061 | **0.048** | 0.084 | 0.069 |
| South Esk | 0.037 | 0.033 | 0.035 | **0.032** | 0.043 | 0.039 |
| Tully | 0.175 | 0.177 | 0.165 | **0.151** | 0.170 | 0.157 |
| Warwick | 0.032 | 0.023 | 0.032 | **0.023** | 0.047 | 0.038 |
| Yatton | 0.062 | 0.056 | 0.053 | **0.048** | 0.061 | 0.057 |

[Figure]

**Figure 2.** Box plots showing the median, $25^{th}$ and $75^{th}$ percentile, end points and outliers that describe the proportion of observations available for gauges within each catchment.

**Table 4.** Average RMSE [mm/h] for simulated streamflow in evaluation periods for each catchment using IDW and OGW rainfall. Values in bold indicate the lowest obtained RMSE.

|  | GR4H | | HBV | | PDM | |
|---|---|---|---|---|---|---|
| Catchment | IDW | OGW | IDW | OGW | IDW | OGW |
| Hurdle Creek | 0.033 | 0.033 | 0.041 | 0.036 | 0.037 | **0.029** |
| Onkaparinga | **0.035** | 0.036 | 0.038 | 0.039 | 0.038 | 0.039 |
| Paddys Flat | 0.083 | 0.103 | 0.082 | **0.077** | 0.093 | 0.080 |
| South Esk | 0.037 | **0.037** | 0.044 | 0.043 | 0.053 | 0.044 |
| Tully | 0.176 | 0.184 | 0.161 | **0.151** | 0.167 | 0.156 |
| Warwick | 0.035 | 0.038 | 0.042 | **0.031** | 0.050 | 0.041 |
| Yatton | **0.068** | 0.075 | 0.069 | 0.072 | 0.068 | 0.078 |

[Figure]

**Figure 3.** The average annual change in estimated areal rainfall for the simulation period between the OGW and IDW areal rainfall estimation methods.

**Table 5.** Average annual change [mm] in rainfall minus evapotranspiration between the IDW and OGW rainfall simulations. A positive change indicates that net model forcings were higher for the OGW simulations.

|  | GR4H | HBV | PDM |
|---|---|---|---|
| Hurdle Creek | 40.0 | 22.9 | 19.8 |
| Onkaparinga | 5.1 | 0.57 | 6.3 |
| Paddys Flat | -13.0 | 3.5 | -33.3 |
| South Esk | 54.5 | -5.3 | 16.6 |
| Tully | 789.3 | 201.8 | 189.2 |
| Warwick | 9.4 | 2.6 | 14.0 |
| Yatton | 115.5 | -11.6 | 22.5 |
| Mean | 143.0 | 60.1 | 33.6 |

[Figure]

**Figure 4.** Observed streamflow and IDW areal rainfall estimates for Paddys Flat.

**Figure 5.** Cumulative areal rainfall estimates obtained for each OGW simulation when compared to the cumulative IDW rainfall areal rainfall estimate for Paddy's Flat. 1-1 reference lines are shown in black. Each colour represents a different split sample.

[Figure]

**Figure 6.** A 1-1 comparison of the average OGW areal rainfall estimates with the IDW areal rainfall estimates for each model for Paddy's Flat.

**Figure 7.** A comparison of the IDW gauge weights, as indicated by a ×, with OGW gauge weights for each split sample simulation for Paddy's Flat. A + indicates a potential outlier.

[Figure]

**Figure 8.** Relative optimized parameters for each rainfall-runoff model using OGW and IDW rainfall estimates for Paddy's Flat. Values of 0 and 1 indicate the lower and upper bounds for each parameter respectively.

[Figure]

**Figure 9.** A representation of each models ability to resolve storage for each split sample and areal rainfall estimation method respectively for Paddy's Flat. Each time series starts at the end of the warm-up period and is adjusted to have the same storage as other samples from the model and rainfall estimation method pair. Each colour represents a different split sample.

[Figure]

**Figure 10.** A comparison of the streamflow simulations for the evaluation periods for each model and areal rainfall estimation method for Paddy's Flat. Events can be distinguished by lines of hysteresis and the observed streamflow. 1-1 reference lines are shown in black.

---

## Author Comment (AC3) · 14 Nov 2019

The remarks made by the Reviewers and the Associate Editor are written in italics, and the replies in normal font and blue color. In the citation of the remarks, the original page, paragraph, table, figure and line numbers have been used. However, in the replies these numbers refer to the numbers in the revised manuscript unless otherwise specified.

**Reviewer #3**

This manuscript presents a new method to weigh available rainfall gauges in a way that maximizes the performance of hydrologic model in terms of streamflow simulation. The method is tested for 3 hydrologic models for streamflow simulation of 7 Australian catchments and its performance is compared to a classic inverse distance weighing (IDW) approach

The proposed method has its merits and was reported to outperform IDW method in the majority of tested cases. However, the proposed method was compared to only one arbitrary selected interpolation method using only one arbitrary selected performance measure of model efficiency. Little discussion was provided on possible model parameter uncertainty and its effect on the performance of the proposed method. Moreover, the Abstract and Conclusion of the manuscript highlight the importance of the proposed method for flood forecast, whereas in the performed analysis the emphasis is on the simulation of the historical streamflow time series. Finally, the Introduction lacks presentation of existing methods on rainfall optimization and correction techniques for hydrologic modeling. Therefore, I think substantial revision of the manuscript is required before it can be accepted for publication. Below I present my detailed comments.

**General comments**

1. According to the Abstract and the Conclusion of the manuscript the purpose of the proposed method is to obtain superior flood forecast. However, in the technical part of the analysis the proposed method is not tested for such purposes, but rather for the simulation of historical streamflow time series. In the description of the catchments, noticeable flood events are described in great detail, but none of the information regarding these flood events is used to test the proposed method. The proposed method can be indeed useful for flood forecasting, but since this was not the scope of the analysis I would suggest authors to modify abstract and conclusions accordingly by reporting the results that are actually the outcome of this study

The abstract conclusion and overall manuscript have been modified to discuss streamflow simulation rather than flood forecasting.

2. The focus of Introduction is largely on issues regarding systematic errors in rainfall measurement and quality control of these measurements. This is an important topic indeed, but in my opinion current Introduction is disconnected from the proposed method and the performed analysis. Additional information about existing methods on optimization/correction of rainfall would be more welcome (e.g., optimization of areal rainfall (e.g., Anctil et al., 2006) or using precipitation correction factor as a calibration parameter of hydrologic models that is often used in data scarce conditions (e.g., Schaefli et al., 2007; Duethmann et al., 2013)).

The sentence below has been added to the manuscript on P2L37.

"Through random sampling Anctil et al (2006) explored the impact that different rainfall gauge weightings had on streamflow forecasts and determined that the best performing combination of rain gauges came from a sub-sample of the available rain gauges."

Further the sentence below has been added to the manuscript on P2L60

"Further Schaefli et al(2007) and Duethmann et al (2013) use precipitation correction factors that are suitable for mountainous or data sparse regions."

3. In the Introduction (Line 44-50) the authors show that existing studies do not give a univocal answer on the question which interpolation method result in better streamflow simulation, since the performance of different interpolation methods for streamflow simulation might vary largely when different models, different performance metrics and different catchment are used. Moreover, the authors highlight that none of the existing studies used all interpolation methods making it difficult to identify the most appropriate method for streamflow simulation. I fully agree with these statements of the authors, but I am surprised that for their own study the authors provided little motivation on the choice of a single baseline interpolation method (i.e., IDW) and a single performance measure (i.e., RMSE) for model evaluation. Moreover, the detailed results were presented only for one out of seven catchments. The differences in performance for different catchments were not discussed in the context of different catchment characteristics and climatic settings.

The point raised is valid and has been given much thought and attention. The paragraph below has been added to the manuscript on P3L73.

"For the development of a robust methodology to optimally interpolate rainfall gauges it is essential to consider a variety of catchments, rainfall-runoff models, rainfall interpolation schemes, simulation metrics, and quality of data. This study develops a methodology to optimally interpolate rainfall gauges for streamflow simulation using a variety of catchments, rainfall runoff models and quality of data. A review of different methods for spatial interpolation of rainfall data by Ly et al. (2013) found that no spatial interpolation methodology provided consistently good streamflow simulations. Consequently streamflow simulations forced by OGW rainfall are benchmarked against those obtained with IDW rainfall and are deemed suitable if there is a consistent improvement noticed. Situations in which the OGW is not appropriate are highlighted. Lastly, the OGW methodology is part of the calibration process. As such using an alternate simulation metric would still yield the OGW for that metric."

4. Little attention was paid to the uncertainties that might arise due to the parameter equifinality when the different precipitation inputs are evaluated by goodness-of-fit of simulated streamflow. This is an important topic in the field of precipitation benchmarking (e.g., Yilmaz et al., 2005; Heistermann and Kneis, 2011). The proposed method is likely to suffer from similar problems. Since Figure 7 only report the average value of calibrated parameters for 6 different spilt samples, it is not possible to say if the proposed method is affected by parametric uncertainties, but large variability of gauge weights among split sample (Figure 6) hints in that direction.

Thank you for point this out. Indeed the introduction of more parameters are likely to worsen the equifinality problem. It is for this reason that the performance in the evaluation period is a focal point. The sentence on P3L66 was altered to

"Comparisons of streamflow forecast skill when forced with varying quantitative precipitation estimates (Heisterman et al.2011), and rain gauge, radar, and satellite based precipitation estimates (Yilmaz et al. 2005), demonstrate that calibration may obscure error and bias in rainfall estimates, making it essential that rainfall estimates and their resulting calibrated parameters be thoroughly evaluated outside of the optimization period."

The sentence on P10L289 was altered to

"Despite the increase in the amount of parameters needed for OGW streamflow simulations, the relatively small change in parameter spread observed for each model and rainfall interpolation method suggests that there is little increase in hydrologic model parameter uncertainty."

The sentence on P11L339 was altered to

"Further research could explore the impact that OGW has on parameter uncertainty and possible interactions between OGW and the inclusion of catchment exchange or bias correction parameters."

**Specific comments**

Abstract Line 6-7: According to Line 210 of the Result Section 15.3% and 7.1% correspond to the improvement when only 5 out of 7 catchments are considered.

The sentence on P1L5 was altered to

"For a selection of 5 Australian catchments this methodology was able to yield improvements of 15.3% and 7.1% in the optimization and evaluation periods respectively."

*Line 38-39: It seems for me that the main focus of the study is developing a new method for rain gauge weighing rather than determining a superior interpolation method since only one existing method (i.e., IDW) was examined.*

The sentence on P2L41 was altered to

"Therefore this study focuses on developing an interpolation method that adequately weights important rainfall gauges."

*Line 57-60: What about modeling approaches when precipitation correction factor is used as calibration parameter. I think it will be advantageous to mention these techniques here to highlight the novelty of the proposed method.*

These were mentioned through the rainfall multiplier methodology applied by Kavetski et al. (2006) and Vrugt et al. (2008). The references suggested by the reviewer makes the literature review more comprehensive. Thank you.

The sentence on P2L60 was altered to

"Further Schaefli et al(2007) and Duethmann et al (2013) use precipitation correction factors that are suitable for mountainous or data sparse regions."

*Line 64-66: Despite what is claimed here I found no analysis of potential influence of catchment characteristics on performance of the proposed method.*

The sentence on P12L341 was added to the conclusion.

"Catchment characteristics were not found to influence the applicability of the OGW methodology."

Section 2: No information on used temperature data is provided, which I assume is necessary to run hydrologic models used in this study.

**Temperature is not used for any of the models.**

*Line 69: What method was applied to derive monthly PET? Is that a better choice to rely on monthly to hourly disaggregated values than to compute a simple temperature based PET from hourly temperature data directly?*

The sentence on P3L92 clarifies the rationale for the disaggregation strategy.

This disaggregation strategy is appropriate since streamflow simulations are relatively insensitive to the temporal resolution of PET (Samain and Pauwels 2013).

*Line 74-75: Here readers are referred to Table 1 for catchment properties. However, Table 1 only provides catchment size. Consider change the wording in these Lines or provide more catchment properties in Table 1.*

Table 1 has been modified to include average annual rainfall and maximum and minimum elevation.

Line 79-97: Some details regarding catchments (e.g., when the worst flood has occurred or number of recent floods) are not relevant for the performed analysis. On the other hand, additional information regarding catchment physiography, such as, elevation range, mean annual precipitation etc. would be useful for understanding the difference among the catchments and will give a chance to put the findings of the study in the context of different hydrologic conditions. Moreover, instead of describing geographical location of the catchments consider providing a map of study catchments. Please indicate the role of possible anthropogenic influence on streamflow simulation in these catchments.

Table 1 has been modified to include average annual rainfall and maximum and minimum elevation and Figure 1 shows a catchment locality map. While there are possible anthropogenic influences this question would be best answered in a separate study that utilizes a longer period of observation and different modelling methodology.

**Line 99-101: Please provide a rationale on selecting these three models. Are any of them used in operational flood forecast?**

The sentence on P6L160 clarifies the rationale for choosing each model. The term flood forecast has been removed as a focus of this paper.

"The hydrologic models used in this study were chosen based on their proven performance, different routing schemes, and complexity."

Line 116: The value of power parameter is chosen arbitrary. Please provide a rationale for this value.

The sentence on P5L137 has been updated to

"p is an arbitrary power parameter taken to be 2 for this study. This is a commonly applied value, used operationally by the BoM, and consistent with liu et al (2018)."

*Line 121-135: This portion is difficult to understand. Consider revising it. Please provide more details on optimization procedure for weights definition. Was it a part of calibration procedure, where all the weights were identified simultaneously with model parameters?*

The sentence on P5L140 clarifies this

"To consider the effects of areal rainfall estimation on streamflow simulations an approach to optimize gauge weights as a part of the hydrologic model calibration process was introduced."

Where do coefficients (e.g., 2.03 and -4.74) in equation 4 come from? Please provide an explanation.

The sentence on P6L154 has been updated to clarify this

"The constants in equation 4 can be modified to allow different parameter spaces be searched for each gauge, that has a rainfall time series which is more or less complete than the surrounding gauges."

Finally, in this section you refer to likelihood function that explained only later in Section 3.3.1 and is specific to the selected optimization algorithm. Can the proposed gauge weighing method be used in the context of optimization algorithms that rely on maximizing objective function instead of likelihood function?

The sentence on P5L145 has been updated to clarify this

"likelihood or objective function"

And

The paragraph on P7L189 has been updated to clarify this

"The optimal weighting of rainfall gauges will produce rainfall specific to the chosen likelihood function and will change if a likelihood function which places priority on low flows is chosen instead of a likelihood function that places priority on high flows. The same can be said if objective functions are maximized instead of likelihood functions."

Section 3.2: Please provide more details regarding set up of hydrologic models (i.e., lumped, semidistributed, distributed).

The sentence was altered for clarity on P6L164

"Each catchment is modelled in a lumped fashion."

*Line 145-146: Please explain in more detail what is "catchment exchange term". Is it one of the calibrating parameters?*

The sentence was altered for clarity on P6L169

"A potential problem with the GR4H model is that the catchment exchange parameter X2 allows for both the import or export of water into the model."

Line 175-179: Why RMSE was chosen as a performance measure instead of common metrics such as NSE or KGE?

Both bias and correlation are considered in RMSE. This is now addressed on P7L200 "The Root Mean Square Error (RMSE) was chosen to evaluate model simulations for widespread applicability to explain simulation trends such as bias, standard deviation, and correlation."

Table 2, Table 3: consider merging Table 2 and 3 to ease comparison between optimization and evaluation periods described in Lines 181-192. Consider change the name of the column "Opt" to OGW to be consistent with the text.

The column opt has been updated to OGW in Table 3 and Table 4. The authors feel that it is easier to compare optimization and evaluation periods using two separate tables and would strongly prefer to keep the tables arranged as is. If absolutely necessary they can be merged.

Line 185-187: I did not understand this sentence.

The sentences were modified on P8L213

"This can happen when the scaling of gauge weightings, as described in 3.1.2, does not allow the optimization process to attain the same gauge weights as the IDW method. A possible scenario in which this could happen occurs when the IDW method inappropriately applies a weighting larger than 55% to a single gauge."

Line 189: Where can the reader see that IDW rainfall for GR4H model had the lowest RMSE for 5 out of 7 catchments? According to Table 2 in optimization period 6 out of 7 catchments had lower RMSE for IDW than for OGW. According to Table 3 for evaluation period none of 7 catchments had smaller RMSE for OGW.

This paragraph is discussing results in the evaluation period and consequently only table X is being referred to. Further there was a typo that is now corrected on P8L216

"Whilst the OGW rainfall estimates generally led to improvements in the evaluation period, there was much less consistency in performance between models. For IDW rainfall the GR4H model had the lowest RMSE for 2 out of the 7 catchments. The best simulations were obtained for 5 out of the 7 catchments when OGW rainfall was used."

Line 190-193 and Line 210: Is it feasible to compare the performance of the best IDW to the best OGW case disregarding of the model used? According to Introduction (Line 48-50) the success of interpolation methods is greatly influenced by the choice of hydrologic models.

P8L220 and P8L239 have been updated to the respective sentences

"If the best IDW simulation for all models is compared to the best OGW for all model simulation an average improvement of 4.3% was observed"

"If the results of these two catchments are not considered, comparing the best OGW simulation for all models to the best IDW simulation for all models led to improvements of 15.3% and 7.1% in the optimization and evaluation periods, respectively."

Line 193-203: Is the proposed method is at all feasible for the conditions when gaps are present in the data? Is it possible that the largest weight will be assigned to the gauge with data gap that covers the evaluation period? In this case it is likely to deteriorate model performance in the evaluation period that could have been compensated by the information from other gauges in the traditional interpolation methods.

The authors intended this to be to point conveyed.

*Line 203-206: Is it possible to define/suggest a minimum rain gauge density required for the successful application of the proposed method?*

P8L235 has been updated to include the following sentence;

"Consequently a minimum gauge density of at least one gauge per 200 km2 is recommended."

Section 4.1: Why Paddys Flat was selected as a case study?

P9L261 has been updated to include the following sentence;

"Paddys Flat was chosen for the case study due to the improvement in streamflow simulation skill in the evaluation period being similar to the mean improvement of streamflow simulation skill in the evaluation period across all 7 catchments. Further, results observed in the Paddys Flat case study are considered to be representative of the OGW methodology."

Figure 3: Is this Figure necessary? The Figure simply shows observed streamflow and IDW interpolated rainfall for the whole study period. It would be more informative if apart from observed discharge it will display discharge simulated by different models with corresponding OGW rainfall for a selected period so that discharge would be actually visible on the Figure.

Figure 4 ties in with figure 6 and figure 10 where comparisons of rainfall and streamflow are given for each model and rainfall estimation methodology.

Line 227: OGW leads to increase of cumulative rainfall for HBV and PDM for all split samples, however for some split samples of GR4H the cumulative rainfall is similar to the IDW rainfall.

The phrase has been reworded on P9L268 to

"demonstrating that the OGW tends to lead to an increase in cumulative rainfall volumes"

Line 233-235: These findings are based on Paddys Flat that according to the Figure 2 did not have large change of rainfall amount compared to IDW. Are these results similar for Tully catchment where the difference between OGW and IDW rainfall amounts was larger than 1000 mm?

Yes these results are similar for the Tully catchment. P9L261 has been updated to include the following sentence;

"Paddys Flat was chosen for the case study due to the improvement in streamflow simulation skill in the evaluation period being of similar to the mean improvement of streamflow simulation skill in the evaluation period across all 7 catchments. Further, results observed in the Paddys Flat case study are considered to be representative of the OGW methodology."

*Line 236, Line 281: The developed method identifies gauges that add value for the streamflow simulation. This manuscript did not investigate the value of identified gauges for the flood forecast.*

Text throughout the manuscript has been updated to focus on streamflow simulation

Figure 7: Instead of average value of parameters for OGW please provide a box plot of parameter values resulting from each split sample to evaluate the stability of parameters. The decimal point on y axis is missing.

Figure 8 has been updated to show the all split sample parameters. The decimal point is now clearer.

Line 244-250, Line 258-260: Since the model parameters were not introduced for each model in the Methods Section, please specify which parameter mentioned here and are responsible for soil moisture, evapotranspiration, percolation, fast flow, base flow, catchment exchange parameter etc. to make interpretation of Figure 7 easier.

A table with model parameters and their ranges is given in Table 2. Interested readers are provided with references to the models.

*Line 272-273: This study does not investigate which model structure is likely to benefit more from inclusion of soil moisture for calibration or assimilation purposes. Consider adding citation or revising this sentence.*

A citation of Li et al (2016) has been added on P1L321.

*Line 269-270, 275-278 and Figure 9: Figure 9 is not appropriate to make any conclusions regarding IDW and OGW performance for different streamflow events. On this Figure individual events cannot be identified unambiguously and it is hard to say which IGW event correspond to which OGW event.*

Consider modifying this Figure by selecting several different streamflow events and showing performance of different gauge weighing methods and models for them.

To help distinguish between events the sentence below has been included on P11L323 and in the caption for Figure 10.

"Events can be distinguished by lines of hysteresis and the observed streamflow."

The authors feel that Figure 10 shows the entirety of the streamflow observations and provides more detail than showing one individual event. It is not feasible to include plots for many different events.

Line 279: Add the results for all study catchments as supplementary material to prove this point.

The sentence has been removed.

Line 287-289: Is it because of inability of GR4H to represent the internal dynamics of the catchments or is it because the catchment exchange parameter accounts and corrects for possible bias present in the input data making adjustment of gauge weight (that in essence also corrects for input data bias) redundant?

The sentence, "Further research could explore the impact that OGW has on parameter uncertainty and possible interactions between OGW and the inclusion of catchment exchange or bias correction parameters." has been added on P11L339. Simulation results presented demonstrate that the adjustment of gauge weight is not redundant. Trial runs fixing the catchment exchange parameter to zero found that the simulation performance still declined. Further simulations using GRKAL which has a more detailed soil moisture layer showed that improvements in skill where observed. The presentation of these results is beyond the scope of the paper.

**Technical corrections**

*Line 23 and 37: Using "QC'd" instead of "quality controlled" is confusing. In general, consider spelling "quality control" instead using QC. It is only used 4 times in the paper.*

**Corrected**

Line 41: "Thiessen polygons" instead of "Theissen's polygons"

**Corrected**

Line 46: Abbreviations OK and PCRR are not used further in the text and therefore can be omitted.

**Corrected**

Line 49: "is" instead of "iss"

**Corrected**

Line 82: Abbreviation SA is redundant as it is not used further in the text

**Corrected**

*Line 87 and 93: Abbreviation NSW is redundant as it is not used further in the text Line 95: "Queensland" instead of "Queenslad"*

**Corrected**

Line 106: "operator" instead of "oeprator"

**Corrected**

Line 107: "an" instead of "on"

Corrected

Figure 1: Decimal point on y axis is shifted

Corrected

Figure 4: Label the different split samples according to the period they were calibrated on

Figure 5 and Figure 9 have been updated

**References**

All suggested references were added.

Anctil, F., Lauzon, N., Andréassian, V., Oudin, L., Perrin, C., 2006. Improvement of rainfall-runoff forecasts through mean areal rainfall optimization. J. Hydrol. 328, 717–725. https://doi.org/10.1016/j.jhydrol.2006.01.016

Duethmann, D., Zimmer, J., Gafurov, a., Güntner, a., Kriegel, D., Merz, B., Vorogushyn, S., 2013. Evaluation of areal precipitation estimates based on downscaled reanalysis and station data by hydrologic modelling. Hydrol. Earth Syst. Sci. 17, 2415–2434. https://doi.org/10.5194/hess-17-2415- 2013

*Heistermann, M., Kneis, D., 2011. Benchmarking quantitative precipitation estimation by conceptual rainfall-runoff modeling. Water Resour. Res. 47, 1–23. https://doi.org/10.1029/2010WR009153*

Schaefli, B., Talamba, D.B., Musy, A., 2007. Quantifying hydrologic modeling errors through a mixture of normal distributions. J. Hydrol. 332, 303–315. https://doi.org/10.1016/j.jhydrol.2006.07.005

Yilmaz, K.K., Hogue, T.S., Hsu, K., Sorooshian, S., Gupta, H. V., Wagener, T., 2005. Intercomparison of Rain Gauge, Radar, and Satellite-Based Precipitation Estimates with Emphasis on Hydrologic Forecasting. J. Hydrometeorol. 6, 497–517. https://doi.org/10.1175/JHM431.1

**Insights from a new methodology to optimize rain gauge weighting for rainfall-runoff models**

Ashley J. Wright1, David E. Robertson2, Jeffrey P. Walker1, and Valentijn R.N. Pauwels1 1Department of Civil Engineering, Monash University, Clayton, Victoria, Australia 2Commonwealth Scientific and Industrial Research Organisation, Clayton, Victoria, Australia

Correspondence: Ashley Wright

**Abstract.** The simulation of streamflow continues to be of significant importance for societies and their economies. Hydrologic modelling of the rainfall-runoff process is essential for the simulation of streamflow. Typically rainfall runoff models use areal rainfall estimates that are based on geographical features. This paper introduces a new methodology to optimally blend the

- 5 weighting of gauges for the purpose of obtaining superior streamflow simulation. For a selection of 5 Australian catchments this methodology was able to yield improvements of 15.3% and 7.1% in optimization and evaluation periods respectively. Catchments with a low gauge density, or an overwhelming majority of gauges with a low proportion of observations available, are not well suited to this new methodology. Models which close the water balance and demonstrate internal model dynamics that are consistent with a conceptual understanding of the rainfall-runoff process yielded consistent improvement in streamflow
- 10 simulation skill.

15

**1 Introduction**

Rainfall-runoff models form the basis of hydrologic simulations for a wide range of applications, extending from forecasting floods through to water resources assessments and the design of water management structures. Hydrologic models generate streamflow simulations from forcing data including rainfall. The quality of any hydrologic simulation is dependent on the availability of high quality estimates of areal rainfall for model optimization and simulation.

For rainfall-runoff models the concept of *garbage in, garbage out* is inescapable. An improper treatment of forcing data leads to a cascade of errors which renders all other modeling techniques ineffective (Kuczera et al., 2010). Rainfall is the primary forcing for models which simulate streamflow and can be derived from in-situ gauges, satellite and ground mounted RADARs, and re-analysis products. The quality of observations, their quality control, and the methods to aggregate rainfall

20 observations all play a key role in the effective identification of hydrologic model parameters (Te Linde et al., 2008) and the assimilation (Vanden-Eijnden et al., 2013) of observed states and outputs. Due to relative accuracy, precision, and the near immediate availability of quality controlled observations, methods that utilize gauge data to derive areal rainfall incident upon a catchment are essential for operational flood forecasts (Li et al., 2016) and are the focus of this study.

The quality of gauge based areal rainfall estimates hinges upon systematic errors, quality control of observations, the design of rain gauge networks, and methodologies to interpolate areal rainfall from the gauge network. In a review of WMO gauge inter-comparison studies, Sevruk et al. (2009) stressed the need to correct for systematic error sources such as wind, evaporation, and shielding. Rodda and Dixon (2012) highlight that systematic errors from wind contribute most significantly towards rainfall undercatch while Pollock et al. (2018) demonstrated that a conventional cylinder shaped rain gauge mounted at 0.5 m can underestimate rainfall measured at the ground by a pit gauge by more than 23% on average. Ouality control of gauged data

- 30 sets is often performed via comparison with re-analysis (Robertson et al., 2015) or RADAR based products (Qi et al., 2016). Rain gauge networks are used at different temporal and spatial scales for meteorological, climatological, and hydrologic purposes. Few rain gauge networks are designed to optimize the ability of hydrologic models to simulate streamflow and flood events (Chacon-Hurtado et al., 2017). Dong et al. (2005) demonstrated that local climatic and geographic conditions play a role in the ability of each gauge to improve hydrologic simulation and that some gauges hinder the successful simulation of
- 35 streamflow. For their study catchment Xu et al. (2013) found that hydrologic simulation skill improved significantly until a gauge density of 0.4 gauges per 1000 km2 was reached and that simulation skill degraded for rain gauge densities larger than 1.4 gauges per km2. Through random sampling Anctil et al. (2006) explored the impact that different rainfall gauge weightings had on streamflow forecasts and determined that the best performing combination of rain gauges came from a sub-sample of the available rain gauges. These studies either indicate that systematic errors have not been sufficiently quality controlled or that the methodology to interpolate areal rainfall from the gauge network does not adequately weight important gauges.

Therefore this study focuses on developing an interpolation method that adequately weights important rainfall gauges. There are a variety of interpolation methods used to obtain an areal rainfall estimate from a gauged network ranging from

simple methods such as the Thiessen's polygons and inverse distance weighting (IDW) methods, to more complex methods such as kriging. In a review of different methods for the spatial interpolation of rainfall data Ly et al. (2013) found no optimal

- 45 spatial interpolation method for operational hydrology, and that interpolation performance often depended on the density of the gauge network, the spatial and temporal resolution of the data, and parameters such as those used by the semi-variogram in kriging. Mair and Fares (2011); Chen et al. (2017), and Liu et al. (2018) compared interpolation methods based on their ability to simulate streamflow, with each study using a different hydrologic model and catchment. In respective order they found that ordinary kriging, principal component regression with residual correction, and IDW interpolation of gauged rainfall data
- 50 yielded superior streamflow simulations. However, none of the studies included all three methodologies. The use of a single but different catchment and single but different hydrologic model for each study makes it difficult to determine the robustness of each methodology. It is hypothesized that the best performing interpolation methods are not optimized for hydrologic simulations and that their success is likely to be greatly influenced by the choice of hydrologic model, chosen catchment, and simulation metric.
- The search for an optimal interpolation of rainfall gauges for hydrologic simulation is essentially an inverse modeling question that asks, how do hydrologists extract optimal rainfall forcings from streamflow simulations? Methods to extract optimal rainfall from streamflow range from those that do not need rainfall forcing such as a cascading linear inversion of runoff (Hino, 1986) to those that require rainfall forcing such as linear inversion (Kirchner, 2009), optimization of storm multipliers (Kavetski et al., 2006; Vrugt et al., 2008), reduction of forcing data (Wright et al., 2017), and the inclusion of
- ancillary data (Wright et al., 2018). Further Schaefli et al. (2007) and Duethmann et al. (2013) use precipitation correction

factors that are suitable for mountainous or data sparse regions. The nature of these studies implies that rainfall obtained from interpolation methods is not *true* rainfall. However, no example is found in the literature which presents a methodology to optimally interpolate rainfall gauge data for hydrologic simulations and provide insights derived therein. To develop an algorithm for the optimization of gauge weights (OGW) that allows the estimation of rainfall which optimizes hydrologic

- 65 simulation skill it is necessary to include additional degrees of freedom in the parameter optimization process. Comparisons of streamflow forecast skill when forced with varying quantitative precipitation estimates (Heistermann and Kneis, 2011), and rain gauge, radar, and satellite based precipitation estimates (Yilmaz et al., 2005), demonstrate that calibration may obscure error and bias in rainfall estimates, making it essential that rainfall estimates and their resulting calibrated parameters be thoroughly evaluated outside of the optimization period. Many hydrologic purists are strong advocates of *Occam's Razor* and
- 70 are, rightfully so, concerned that adding hydrologic parameters will result in over-fitting of hydrologic models. Keeping this in mind the value of additional parameters should be determined through the ability of those parameters to provide improved solutions for hydrologic simulations in either an evaluation or forecast period.

For the development of a robust methodology to optimally interpolate rainfall gauges it is essential to consider a variety of; catchments, rainfall-runoff models, rainfall interpolation schemes, simulation metrics, and quality of data. This study develops

- 75 a methodology to optimally interpolate rainfall gauges for streamflow simulation using a variety of catchments, rainfall runoff models and quality of data. A review of different methods for the spatial interpolation of rainfall data by Ly et al. (2013) found that no spatial interpolation methodology consistently provided the best streamflow simulations. Consequently streamflow simulations forced by OGW rainfall are benchmarked against those obtained with IDW rainfall and are deemed suitable if there is a consistent improvement noticed. Situations in which the OGW is not appropriate are highlighted. Lastly, the OGW methodology is part of the optimization process. As such using an alternate simulation metric would still yield the OGW for
- that metric.

This paper contributes to the hydrologic sciences by developing a novel OGW method which is used in the estimation of areal rainfall for hydrologic simulations and providing insights on the potential influences that catchment characteristics and the choice of hydrologic model pose for areal rainfall estimates based on gauge interpolation.

**85 2 Data set**

With locations shown in Figure 1 this study uses seven Australian catchments that are susceptible to major flooding, and having different rainfall gauge densities, climatology, area, response times, flow characteristics, and seasonality of rainfall. Further, data availability was also an important factor in choosing the catchments. Data are collected for the 7 year period beginning  $1^{st}$  of January 2007 and ending  $31^{st}$  of December 2013. With the exception being that rainfall in the Onkaparinga and South

90 Esk catchments was first recorded on 1st of November 2007. Rainfall and potential evapotranspiration (PET) data are used to force the models. Monthly PET data from the Australian Water Availability Project (AWAP) (Raupach et al., 2009, 2012) were used at and disaggregated to a uniform hourly rate for each month. This disaggregation strategy is appropriate since streamflow simulations are relatively insensitive to the temporal resolution of PET (Samain and Pauwels, 2013). Both hourly rainfall and

streamflow data (Bureau of Meteorology, 2019b) were obtained from the Bureau of Meteorology (BoM) with the catchments

- being delineated using the Australian geofabric data set (Bureau of Meteorology, 2019a). All rainfall gauges within 10 km of 95 the catchment boundary were considered. A summary of the catchment properties is given in Table 1, whilst an indication of the available observations for each rain gauge in each catchment is given in Figure 2. When no gauge within the catchment had an observation, rainfall is assumed to be zero. Ephemeral streams were defined as those having zero flows in > 4% of their records (Bennett et al., 2016), and Köppen Geiger climate classifications were taken from Peel et al. (2007). Brief summaries of each catchments main characteristics are as follows: 100
- - Hurdle Creek at Bobinawarrah has a small catchment that drains to an ephemeral tributary stream of the frequently flooded Ovens River in north-eastern Victoria. It has a temperate climate and a near uniform seasonal rainfall distribution.
  - Onkaparinga River upstream of Hahndoorf dissipator lies near the head of the flood prone Onkaparinga River catchment. The ephemeral stream is located within a temperate climate in south-eastern South Australia and has winter rains with drier summers.
  - South Esk River at Llewellyn is part of a river system with a long history of flooding. The intermittent stream is located in north-eastern Tasmania and belongs to a temperate climate that has a near uniform seasonal rainfall distribution.
  - Clarence River at Paddys Flat is an intermittent stream at the headwaters of a flashy catchment that observed its worst flood in January of 2013. The catchment lies in the eastern part of New South Wales just south of the Oueensland border in a temperate climate that has winter rains and drier summers.
  - Tully River at Euramo is a perennial stream that has been consistently subject to moderate floods over the last two decades. It lies within a tropical monsoon climate in far north Queensland that has the bulk of its rainfall over summer.
  - Condamine River at Warwick is an ephemeral stream that contributes to the slow moving Condamine River. The Condamine River has seen some of the countries worst flooding, having had 3 majors floods in the last decade alone. The catchment lies north of the New South Wales border in the eastern part of Queensland in a temperate climate that has winter rains and drier summers.
  - Isaac River at Yatton is an ephemeral stream that lies within a hot semi-arid climate in north Queensland that is subject to large summer rainfall events. The Isaac River is a tributary of the Mackenzie and Fitzroy Rivers and consequently plays a vital role in flood warning. The catchment of Yatton has been subject to 3 major floods in the last decade.

**3 Methods 120**

105

110

115

In this study, the effects of two different methods of estimating areal rainfall on streamflow simulation performance are compared using three different hydrologic models. Here the methods of estimating areal rainfall are introduced, followed by a description of the hydrologic modelling approach and techniques for evaluating streamflow simulation performance.

4

**3.1 Areal rainfall estimation**

130

125 In essence the estimation of areal rainfall ri, at time step i, from rainfall gauges involves adequately estimating the weighting of gauges and is determined by

$$\mathbf{r}_{i} = \sum_{j=1}^{n} (\mathbf{r}_{i,j}. \times \mathbf{w}_{i,j}), \tag{1}$$

where  $r_{i,j}$  and  $w_{i,j}$  are the rainfall volume and gauge weighting for each timestep i and gauge j respectively. The  $\times$  operator indicates element by element multiplication. For each timestep where a gauge had an observation, the gauge weights  $w_{i,j}$  are automatically scaled to unity by

$$\mathbf{w}_{i,j} = \mathbf{f}_{i,j} \cdot \mathbf{\times} \mathbf{w}_{j} \cdot / (\mathbf{f}_{i,j} \times \mathbf{w}_{j}), \tag{2}$$

where  $w_j$  is the gauge weight for each location and  $f_{i,j}$  is the binary filter array which describes if a gauge had an observation at each time step.

**3.1.1 Inverse distance weighting**

135 In contrast to the OGW method, both gauge weights and areal rainfall estimation are calculated prior to the estimation of model parameters. Gauge weights were calculated by

$$w_j = 1/d_j^{p}, \tag{3}$$

where  $d_j$  is the distance from gauge j to the catchment centroid and p is an arbitrary power parameter taken to be 2 for this study. This is a commonly applied value, used operationally by the BoM, and consistent with Liu et al. (2018).

**140 3.1.2 Optimization of gauge weights**

Methods such as IDW do not consider the hydrologic model performance in estimating areal rainfall. To consider the effects of areal rainfall estimation on streamflow simulations an approach to optimize gauge weights as a part of the hydrologic model optimization process is introduced. To allow for the identification of optimal gauge weights via a parameter estimation algorithm, maximum and minimum parameter bounds need to be identified. During the optimization process, if weighting for a

- 145 gauge were able to become zero then data for that gauge is effectively discarded. If this were able to occur it is possible that the chosen likelihood or objective function becomes maximized by discarding multiple gauges and assigning an unrealistic weight to one gauge. In this situation the hydrologic model parameters could be over-fitted. Over-fitting is defined to occur when the addition of parameters leads to improved performance in the optimization periods and decreased performance in the evaluation periods. To avoid this gauge weights, wminj, were assigned a minimum un-scaled value of one. It is then logical that gauges which have observations available for a larger proportion of time to potentially have larger gauge weightings. To ensure that
- the gauges with less observations contribute less to areal rainfall estimates, this maximum un-scaled gauge weighting reduced

according to an appropriate negative exponential. The maximum allowable un-scaled gauge weightings  $w_j^{\max}$  that could be explored were calculated as

 $\mathbf{w}_{i}^{\max} = \max(1.01, (n-1) \times 0.55/0.45 \times (1 - 2.03 \times e^{-4.74 \times \mathbf{p}_{j}})), \tag{4}$

155 where n is the total number of gauges used and  $p_j$  is the proportion of observations available for gauge j. The constants in equation 4 can be modified to allow different parameter spaces to be searched for each gauge, that has a rainfall time series which is more or less complete than the surrounding gauges. Further,  $w_j^{max}$  were restricted to values that ensured a maximum scaled weighting of 0.55 if that gauge had all observations available. Similar to the IDW method, any alterations to forcing data via re-analysis products or inclusion of additional gauges will accordingly require the optimization process to be revised.

**160 3.2 Hydrologic models**

The hydrologic models used in this study were chosen based on their proven performance, contrasting routing schemes, and complexity. A description of the parameters and their ranges can be found in Table 2. Both the modèle de Génie Rural à 4 paramètres Horaire (GR4H) and Hydroliska Byråns Vattenbalansavdelning (HBV) model use unit hydrograph (UH) routing whilst the Probability Distributed Model (PDM) uses a dynamic storage based routing approach. Both HBV and PDM close the water balance whilst GR4H does not (Perrin et al., 2003). Each catchment is modelled in a lumped fashion.

**3.2.1 GR4H**

165

The GR4H is an hourly application of the modèle de Génie Rural à 4 paramètres Journalier (GR4J) model (Perrin et al., 2003) which utilizes two state variables, soil water storage and routing water storage. The model is mathematically parsimonious and consists of 4 parameters which, along with the state variables, describe evapotranspiration, percolation, and both slow and fast

170 runoff. A potential problem with the GR4H model is that the catchment exchange parameter X2 allows for both the import or export of water into the model. Thus the water balance is not closed and potential biases in observation data are not able to be corrected.

**3.2.2 PDM**

The PDM as described by Moore (2007) consists of a set of functions used to describe various hydrologic systems. The model consists of four states, a cascade of two linear stores which are used to describe surface runoff, a linear store used to describe subsurface flow and a catchment based soil moisture store that is considered to consist of soil moisture stores with varying capacities that are able to be represented by a Pareto distribution. The PDM uses 9 model parameters and routing is a dynamic storage based process.

**3.2.3 HBV**

180 Of the many HBV model (Lindström et al., 1997) variants, this paper uses the version developed by Matgen et al. (2006) and used by Pauwels and De Lannoy (2015). The model consists of three state variables, a soil reservoir, a slow reservoir,

and a fast reservoir. The 11 parameters, 3 state variables, and subsequent governing equations are used to calculate the actual evapotranspiration, infiltration, effective rainfall, percolation, and proportion of effective rainfall that enters the fast and slow reservoirs, outflow from each reservoir, and UH based routing.

**185 3.3 Parameter estimation**

The estimation of model and OGW parameters was performed simultaneously using the Differential Evolution Adaptive Metropolis algorithm with past state sampling and snooker updating (DREAMZS) of Vrugt (2016). The default settings outlined in Vrugt (2016) were used. DREAMZS finds the posterior parameter distribution which maximizes a chosen likelihood function.

**190 3.3.1 Likelihood function**

The optimal weighting of rainfall gauges will produce rainfall specific to the chosen likelihood function and will change if a likelihood function which places priority on low flows is chosen instead of a likelihood function that places priority on high flows. The same can be said if objective functions are maximized instead of likelihood functions. For this study a Gaussian likelihood function (Thiemann et al., 2001) is used.

**195 3.4 Evaluation strategy**

The evaluation of optimized parameters and gauge weightings was carried out via split sample testing. A warm-up period of one year beginning  $1^{st}$  of January 2007 and ending  $31^{st}$  of December 2008 was used. In the Onkaparinga and South Esk catchments a warm-up period of two months was used. The skill of streamflow simulations was then evaluated for the six year period beginning  $1^{st}$  of January 2008 and ending  $31^{st}$  of December 2013. The optimization and evaluation periods were 5 and 1 years long respectively. The optimization and evaluation periods were alternated until all 6 years of evaluation data had been

200

used for evaluation. This resulted in a total of 6 split samples for each catchment. The Root Mean Square Error (RMSE) was chosen to evaluate model simulations for widespread applicability to explain

simulation trends such as bias, standard deviation, and correlation. RMSE is given by

$$RMSE = \sqrt{\frac{\sum_{i=1}^{t} (q_i^s - q_i^o)^2}{t}},$$
(5)

[revised manuscript text omitted]

- 245 model. If some models are not well suited to be used with OGW rainfall estimates then it is imperative to know which elements of the model structure contribute to this decrease in skill. The average annual change in rainfall and net forcing, where net forcing are defined as rainfall minus evapotranspiration, between IDW and OGW simulations, are shown in Figure 3 and Table 5 respectively. According to the IDW rainfall interpolation the Tully catchment does not observe enough rainfall to explain the observed streamflow. Consequently, it is likely that the OGW method compensates for this by weighting gauges that observe
- 250 the highest rainfall for the Tully. OGW rainfall is larger than IDW rainfall for all hydrologic models and catchments, except Yatton. It was also common for both HBV and PDM to obtain larger OGW rainfall estimates than GR4H. This trend is in stark contrast to the pattern in the change of net forcing. The mean change in net forcing across catchments was significantly lower for HBV and PDM than it was for GR4H, indicating that the internal dynamics of the HBV and PDM adapt to a change in rainfall by altering evapotranspiration while GR4H does not.
- It is common for hydrologists to link the improved simulation performance and increase in the number of parameters with over-fitting. The consistent improvement in simulation skill in the evaluation periods for the PDM and HBV model demonstrates that over-fitting does not occur for these models. However, the consistent decrease in simulation skill in the evaluation periods for the GR4H model demonstrates that over-fitting did occur for this model. By testing different gauge weightings the OGW methodology takes greater advantage of the available rainfall time series when compared to the IDW methodology. As such it is hypothesized that the GR4H model is not able to take advantage of the additional data and is subject to over-fitting.

**4.1 Case Study - Paddys Flat**

This case study was designed to develop a deeper understanding of the impact that the OGW and subsequent areal rainfall estimates had on the hydrologic model simulations and internal dynamics. Paddys Flat was chosen for the case study due to the improvement in streamflow simulation skill in the evaluation period being similar to the mean improvement of streamflow simulation skill in the evaluation period being similar to the mean improvement of streamflow are considered to be representative of the OGW methodology. Results and their associated discussion are presented sequentially to describe the impacts on rainfall, model parameters, the water balance, and streamflow. The observed streamflow and IDW areal rainfall estimate for Paddys Flat can be seen in Figure 4.

The comparison of the cumulative rainfall volumes obtained for the split sample simulations for each model at Paddys Flat is shown in Figure 5 demonstrating that the OGW tends to lead to an increase in cumulative rainfall volumes. Each split sample obtained through model evaluation is represented by a different line. The cumulative rainfall volumes estimated by GR4H are less than those estimated by HBV and PDM. Each split sample produced different behavior according to the data set it was calibrated against. It is expected that results would become more homogenous when longer data sets are used for optimization. However, it is necessary to determine if there is a tendency for low or high magnitude rainfall observations to be estimated

275 differently by the OGW or IDW interpolation methods.

Plots of the OGW estimation of areal rainfall obtained by each model versus the IDW estimation of areal rainfall in Figure 6 reveal that, when compared to the IDW interpolation method, the OGW methodology did not have a tendency to predict greater rainfall volumes for small or large magnitude rainfall observations. The larger cumulative rainfall volumes observed by the OGW method were therefore a result of a slight tendency to estimate greater rainfall volumes for both small and large magnitude rainfall observations. The impact each gauge has on the estimation of areal rainfall was then explored.

280

305

Hydrologic simulation skill based OGWs allows gauges which add value to evaluation periods or forecasts to be identified. The gauge weightings determined by the IDW interpolation method and OGW method for each split sample and model are shown in Figure 7. Each model selected similar gauges and weightings for the majority of gauges. There was however one or two gauges which only one model assigned weight to. Therefore the results from the analysis of rainfall estimates obtained

- 285 through the OGW method do not indicate that GR4H, HBV, and PDM require model specific rainfall forcings, and nor do they give sufficient reasoning as to why the OGW simulations decrease in simulation skill for GR4H and improve in simulation skill for HBV and PDM. It is therefore likely that some difference in model parameters and subsequent impact on internal model dynamics caused the increase/decrease in skill.
- To demonstrate the change or lack of change in model parameters that were observed for each rainfall estimation method, 290 the Maximum a Posteriori (MAP) parameters obtained for each split sample simulation are shown for the OGW and IDW simulations for Paddys Flat in Figure 8. Despite the increase in the amount of parameters needed for OGW streamflow simulations, the relatively small change in parameter spread observed for each model and rainfall interpolation method suggests that there is little increase in the hydrologic model parameter uncertainty. GR4H parameters did not noticeably alter whilst parameters that influence soil moisture content, evapotranspiration, and percolation change significantly for both HBV and PDM. This lack of change in parameters for GR4H indicates that the model is not taking advantage of the additional data. Further, parameters
- influencing fast flow and baseflow change for the HBV model. The change in these specific parameters indicate that internal model dynamics are sensitive to changes in forcing data for both HBV and PDM but not for GR4H. This presents a basis as to why rainfall minus evapotranspiration change much less for both HBV and PDM than GR4H.

The water balance is given as

300
$$s_{\rm i} = \sum_{m=1}^{i} (r_{\rm m} - q_{\rm m} - e_{\rm m}),$$
 (6)

where  $s_i$ ,  $r_m$ , and  $e_m$  are the catchment storage, rainfall forcing, and evapotranspiration at time steps i and m respectively. It should be noted that for GR4H a catchment exchange process that either abstracts or provides additional water may occur. Figure 9 shows the water balance for the split sample simulations for the OGW and IDW methods for each model at Paddys Flat. Again each split sample produced different behavior according to the data set it was calibrated against and it is expected that results would become more homogenous when longer data sets are used for optimization. After the warm-up period of

one year, simulations for both the HBV and PDM models reached a point of equilibrium in which a rise or decline in  $s_i$  was eventually countered by a subsequent and opposing rise or decline in  $s_i$ . This trend is the cornerstone of a model that is able to close the water balance. A non-zero positive or negative catchment exchange parameter in GR4H led to simulations which do not close the water balance. This is the reason why si gradually increased for GR4H. Referring back to Figure 8 it can be seen

310

that the catchment exchange parameter was marginally above the mean of the minimum and parameter bounds. This mean is zero and explains why there is only a gradual departure from zero.

The IDW storage profiles and the OGW storage profiles obtained using GR4H remained inconsistent with the current conceptual understanding of the rainfall-runoff process. The storage profiles for GR4H have different trajectories for each split sample which became more dispersed with the inclusion of OGW rainfall estimates. In contrast, both HBV and PDM demon-

315 strated storage profiles that tend towards an equilibrium. The storage profiles of HBV and PDM for each split sample are more similar to each other than when the IDW rainfall estimates were used. Further, the storage profiles obtained using the HBV and PDM were remarkably similar for all split samples regardless of whether or not the IDW or OGW rainfall estimates were used. As such both, HBV and PDM demonstrate improved internal model dynamics which are consistent with a conceptual understanding of the rainfall-runoff process. Lastly, the ability of each model to represent different streamflow events was analyzed.

The OGW estimation of areal rainfall enabled both HBV and PDM to simulate internal dynamics with improved consistency. Models that have more consistency in their representation of internal dynamics are better positioned to benefit from the inclusion of soil moisture data for calibration and/or assimilation purposes (Li et al., 2016). Similar benefits are likely to be observed when updating internal states through the assimilation of observed streamflow. The relative impact of the OGW methodology

- 325 on streamflow, when compared to the IDW methodology, can be observed in Figure 10. Events can be distinguished by lines of hysteresis and the observed streamflow. For all three models there are a number of flood events that are not adequately simulated. It is hypothesized that this occurs as a result of insufficient rainfall being observed to simulate the observed streamflow. It is clear that for larger magnitude events the OGW methodology brought the streamflow simulations closer to observations for the HBV and PDM throughout the evaluation period. Conversely, the use of OGW rainfall areal rainfall estimates with GR4H
- 330 led to marginally improved streamflow simulations for one event and significantly worse streamflow simulations for another event.

**5 Conclusions**

This study developed a methodology to optimally weight rainfall gauges such that improved streamflow simulation skill could be obtained for three different conceptual hydrologic models. The OGW methodology developed was tested on seven Australian catchments and a comparison of streamflow simulations obtained using the IDW and OGW methods demonstrated an improvement in streamflow RMSE of 15.3% and 7.1% in the optimization and evaluation periods respectively, for catchments that have multiple rainfall gauges with observations available > 50% of the time and more than one rainfall gauge every 200 km2. The methodology did not work equally well for the three hydrologic models chosen. Improvements in evaluation periods were only noticed for the PDM and HBV hydrologic models, and not for the GR4H hydrologic model. The most likely explanation for this is the inability of the GR4H model to represent internal dynamics that are consistent with a conceptual understanding of the rainfall-runoff process. Further research could explore the impact that OGW has on parameter uncertainty and possible interactions between OGW and the inclusion of catchment exchange or bias correction parameters. Catchment characteristics were not found to influence the applicability of the OGW methodology. This methodology opens new possibilities for model evaluation, understanding of forcing uncertainty, and data assimilation studies in hydrology.

345 *Author contributions*. AW conducted the experimental work, contributed towards the theory and wrote the paper. JW assisted in the writing process. VP and DR contributed towards the theory and assisted in the writing process.

*Acknowledgements.* The authors would like to extend their gratitude to the anonymous reviewers for their comments and recommendations. This work was supported by the Multi-modal Australian Sciences Imaging and Visualisation Environment (MASSIVE) (http://www.massive.org.au).

[revised manuscript text omitted]

**Table 1.** Catchment area, number of rainfall gauges, the % of time that no gauge had an observation for each of the study catchments, average annual IDW rainfall over the simulation period, and minimum and maximum elevation. It should be noted that IDW rainfall is not regarded as the truth. Statistics shown are for the collected rainfall record beginning  $1^{st}$  of January 2007 and ending  $31^{st}$  of December 2013, except for in the Onkaparinga and South Esk catchments where rainfall was first recorded on  $1^{st}$  of November 2007.

| Catchment   | Number of | % of time | Catchment            | Annual    | Minimum   | Maximum   |
|-------------|-----------|-----------|----------------------|-----------|-----------|-----------|
|             | gauges    | no gauge  | area $\mathrm{km}^2$ | IDW rain- | elevation | elevation |
|             |           | works     |                      | fall [mm] | [m]       | [m]       |
| Hurdle      | 12        | 14.6      | 156                  | 582.6     | 165.3     | 1062.4    |
| Creek       |           |           |                      |           |           |           |
| Onkaparinga | 21        | 12.5      | 208                  | 485.4     | 179.0     | 616.6     |
| Paddys Flat | 27        | 2.6       | 3112                 | 765.1     | 168.6     | 1300.6    |
| South Esk   | 13        | 6.902     | 2289                 | 616.2     | 177.5     | 1527.3    |
| Tully       | 7         | 4.1       | 1390                 | 2133.2    | 6.7       | 1198.6    |
| Warwick     | 21        | 5.3       | 1379                 | 502.1     | 446.2     | 1361.7    |
| Yatton      | 7         | 3.6       | 19639.5              | 1015.8    | 99.0      | 1084.9    |

Vrugt, J. A., ter Braak, C. J. F., Clark, M. P., Hyman, J. M., and Robinson, B. A.: Treatment of input uncertainty in hydrologic modeling: Doing hydrology backward with Markov chain Monte Carlo simulation, Water Resour. Res., 44, https://doi.org/10.1029/2007WR006720, 2008.

- Wright, A., Walker, J., and Pauwels, V.: Estimating rainfall time series and model parameter distributions using model data reduction and inversion techniques, Water Resour. Res., 53, https://doi.org/10.1002/2017WR020442, 2017.
- Wright, A. J., Walker, J. P., and Pauwels, V. R. N.: Identification of Hydrologic Models, Optimized Parameters, and Rainfall Inputs Consistent with In Situ Streamflow and Rainfall and Remotely Sensed Soil Moisture, J. Hydrometeorol., 19, 1305–1320, https://doi.org/10.1175/JHM-D-17-0240.1, 2018.

430 https://doi.org/10.1175/JHM-D-17-0240.1, 2018.

425

435

- Xu, H., Xu, C.-Y., Chen, H., Zhang, Z., and Li, L.: Assessing the influence of rain gauge density and distribution on hydrological model performance in a humid region of China, J. Hydrol., 505, 1–12, https://doi.org/10.1016/j.jhydrol.2013.09.004, 2013.
- Yilmaz, K. K., Hogue, T. S., Hsu, K.-L., Sorooshian, S., Gupta, H. V., and Wagener, T.: Intercomparison of rain gauge, radar, and satellitebased precipitation estimates with emphasis on hydrologic forecasting, J. Hydrometeorol., 6, 497–517, https://doi.org/10.1175/JHM431.1, 2005.
- Zeng, Q., Chen, H., Xu, C.-Y., Jie, M.-X., Chen, J., Guo, S.-L., and Liu, J.: The effect of rain gauge density and distribution on runoff simulation using a lumped hydrological modelling approach, J. Hydrol., 563, 106–122, https://doi.org/10.1016/j.jhydrol.2018.05.058, 2018.

| Model | Parameter                   | Description                                                                | Range                        | Unit                         |
|-------|-----------------------------|----------------------------------------------------------------------------|------------------------------|------------------------------|
| GR4H  | X1                          | Production store capacity                                                  | 1 - 3000                     | mm                           |
|       | X2                          | Catchment exchange                                                         | -27 - 27                     | mm                           |
|       | X3                          | Routing store capacity                                                     | 1 - 660                      | mm                           |
|       | X4                          | length of unit hydrograph                                                  | 1 - 240                      | hr                           |
| HBV   | $\lambda$                   | Parameter for evapotranspiration                                           | $10^{-2}$ - 3                | -                            |
|       | $S_{\max}$                  | Capacity of soil reservoir                                                 | $1 - 10^{3}$                 | mm                           |
|       | b                           | Parameter for infiltration                                                 | $10^{-2}$ - 3                | -                            |
|       | Pe                          | Percolation parameter                                                      | $10^{-}2 - 50$               | mm/h                         |
|       | lpha                        | Fast reservoir parameter                                                   | $10^{-2}$ - 3                | -                            |
|       | $\beta$                     | Percolation parameter                                                      | $10^{-2}$ - 3                | -                            |
|       | $S2_{max}$                  | Capacity of fast reservoir                                                 | 1 - 2000                     | mm                           |
|       | $\kappa_2$                  | Parameter for outflow from fast reservoir                                  | $10^{-1} - 10^2$             | $\mathrm{mm}^{3}/\mathrm{h}$ |
|       | $\psi$                      | Parameter for outflow from fast reservoir                                  | $10^{-1}$ - 3                | -                            |
|       | $\kappa_1$                  | Parameter for outflow from slow reservoir                                  | $10^{-3}$ - 1                | $\mathrm{mm}^2/\mathrm{h}$   |
|       | UH                          | length of unit hydrograph                                                  | 1 - 240                      | h                            |
| PDM   | $\mathrm{C}_{\mathrm{max}}$ | Maximum store capacity                                                     | 1 - 500                      | mm                           |
|       | b                           | Parameter that controls spatial variability of $\mathrm{C}_{\mathrm{max}}$ | $10^{-3}$ - 1.8              | -                            |
|       | $\mathbf{b}_{\mathbf{e}}$   | Actual evaporation exponent                                                | 0.1 - 5                      | -                            |
|       | $\mathbf{b}_{\mathbf{g}}$   | Recharge function exponent                                                 | 0.2 - 6.7                    | -                            |
|       | $k_{\rm b}$                 | Baseflow constant                                                          | 1 - 2000                     | $\rm h/mm^2$                 |
|       | $\mathrm{C}_{\min}$         | Minimum store capacity                                                     | $(0-1)\times C_{\max}$       | mm                           |
|       | $\mathbf{S}_{\mathbf{t}}$   | Soil tension storage capacity                                              | $(0 - 1) \times C_{\max}$    | mm                           |
|       | $\kappa_1$                  | Time constant for linear reservoir                                         | 1 - 300                      | h                            |
|       | $\kappa_2/$                 | Time constant for linear reservoir                                         | $(10^{-6} - 100) \times k_1$ | h                            |

Table 2. Parameters and ranges used for the optimization of hydrologic models .

Figure 1. The location of catchments used in this study.

**Table 3.** Average RMSE [mm/h] for simulated streamflow in the optimization periods for each catchment using IDW and OGW rainfall.

 Values in bold indicate the lowest obtained RMSE.

|              | GR4H  |       | H     | BV    | PDM   |       |  |
|--------------|-------|-------|-------|-------|-------|-------|--|
| Catchment    | IDW   | OGW   | IDW   | OGW   | IDW   | OGW   |  |
| Hurdle Creek | 0.033 | 0.031 | 0.032 | 0.028 | 0.033 | 0.028 |  |
| Onkaparinga  | 0.040 | 0.039 | 0.037 | 0.037 | 0.039 | 0.039 |  |
| Paddys Flat  | 0.068 | 0.053 | 0.061 | 0.048 | 0.084 | 0.069 |  |
| South Esk    | 0.037 | 0.033 | 0.035 | 0.032 | 0.043 | 0.039 |  |
| Tully        | 0.175 | 0.177 | 0.165 | 0.151 | 0.170 | 0.157 |  |
| Warwick      | 0.032 | 0.023 | 0.032 | 0.023 | 0.047 | 0.038 |  |
| Yatton       | 0.062 | 0.056 | 0.053 | 0.048 | 0.061 | 0.057 |  |